# Obesity-linked suppression of membrane-bound *O*-acyltransferase 7 (MBOAT7) drives non-alcoholic fatty liver disease

Robert N Helsley[1,2], Venkateshwari Varadharajan[1], Amanda L Brown[1], Anthony D Gromovsky[1], Rebecca C Schugar[1], Iyappan Ramachandiran[1], Kevin Fung[1], Mohammad Nasser Kabbany[1], Rakhee Banerjee[1], Chase K Neumann[1], Chelsea Finney[1], Preeti Pathak[1], Danny Orabi[1], Lucas J Osborn[1], William Massey[1], Renliang Zhang[1], Anagha Kadam[1], Brian E Sansbury[3], Calvin Pan[4,5,6], Jessica Sacks[7], Richard G Lee[8], Rosanne M Crooke[8], Mark J Graham[8], Madeleine E Lemieux[9], Valentin Gogonea[10], John P Kirwan[7], Daniela S Allende[11], Mete Civelek[12], Paul L Fox[1], Lawrence L Rudel[13], Aldons J Lusis[4,5,6], Matthew Spite[3], J Mark Brown[1]*

[1]Department of Cardiovascular and Metabolic Sciences, Cleveland Clinic, Cleveland, United States; [2]Department of Internal Medicine, University of Cincinnati, Cincinnati, United States; [3]Center for Experimental Therapeutics & Reperfusion Injury, Department of Anesthesiology, Perioperative and Pain Medicine, Brigham and Women's Hospital, Harvard Medical School, Boston, United States; [4]Department of Medicine, University of California, Los Angeles, Los Angeles, United States; [5]Department of Microbiology, University of California, Los Angeles, Los Angeles, United States; [6]Department of Human Genetics, University of California, Los Angeles, Los Angeles, United States; [7]Department of Pathobiology, Cleveland Clinic, Cleveland, United States; [8]Cardiovascular Group, Antisense Drug Discovery, Ionis Pharmaceuticals, Inc, Carlsbad, United States; [9]Bioinfo, Plantagenet, Canada; [10]Department of Chemistry, Cleveland State University, Cleveland, United States; [11]Department of Anatomical Pathology, Cleveland Clinic, Cleveland, United States; [12]Department of Biomedical Engineering, University of Virginia, Charlottesville, United States; [13]Department of Pathology, Section on Lipid Sciences, Wake Forest University School of Medicine, Winston-Salem, United States

*For correspondence: brownm5@ccf.org

**Abstract** Recent studies have identified a genetic variant rs641738 near two genes encoding membrane bound *O*-acyltransferase domain-containing 7 (*MBOAT7*) and transmembrane channel-like 4 (*TMC4*) that associate with increased risk of non-alcoholic fatty liver disease (NAFLD), non-alcoholic steatohepatitis (NASH), alcohol-related cirrhosis, and liver fibrosis in those infected with viral hepatitis (Buch et al., 2015; Mancina et al., 2016; Luukkonen et al., 2016; Thabet et al., 2016; Viitasalo et al., 2016; Krawczyk et al., 2017; Thabet et al., 2017). Based on hepatic expression quantitative trait loci analysis, it has been suggested that *MBOAT7* loss of function promotes liver disease progression (Buch et al., 2015; Mancina et al., 2016; Luukkonen et al., 2016; Thabet et al., 2016; Viitasalo et al., 2016; Krawczyk et al., 2017; Thabet et al., 2017), but this has never been formally tested. Here we show that *Mboat7* loss, but not *Tmc4*, in mice is sufficient to promote the progression of NAFLD in the setting of high fat diet. *Mboat7* loss of function is associated with accumulation of its substrate lysophosphatidylinositol (LPI) lipids, and direct administration of LPI promotes hepatic inflammatory and fibrotic transcriptional changes in an *Mboat7*-dependent

manner. These studies reveal a novel role for MBOAT7-driven acylation of LPI lipids in suppressing the progression of NAFLD.

DOI: https://doi.org/10.7554/eLife.49882.001

## Introduction

Non-alcoholic fatty liver disease (NAFLD) is an increasingly common condition that affects roughly one-third of adults in the United States (*Cohen et al., 2011*; *Machado and Diehl, 2016*; *Rinella and Sanyal, 2016*; *Wree et al., 2013*). The expansion of adipose tissue in obese individuals is strongly associated with the development of NAFLD, yet mechanisms linking obesity to NAFLD and more advanced forms of liver disease such as NASH and cirrhosis are not well understood (*Cohen et al., 2011*; *Machado and Diehl, 2016*; *Rinella and Sanyal, 2016*; *Wree et al., 2013*). Genome-wide association studies (GWAS) provide a powerful unbiased tool to identify new genes and pathways that are involved in human disease, allowing for pinpoint accuracy in identification of new drug targets. This is exemplified by the recent success story of human genetic studies leading to rapid development of monoclonal antibodies targeting proprotein convertase subtilisin/kexin type 9 (PCSK9) for hyperlipidemia and cardiovascular disease (*Hess et al., 2018*). Within the last two years, several independent GWAS studies have identified a liver disease susceptibility locus (rs641738) within a linkage-disequilibrium block that contains genes encoding *MBOAT7* and *TMC4* (*Buch et al., 2015*; *Mancina et al., 2016*; *Luukkonen et al., 2016*; *International Liver Disease Genetics Consortium et al., 2016*; *Viitasalo et al., 2016*; *Krawczyk et al., 2017*; *Thabet et al., 2017*). Strikingly, the rs641738 variant is associated with all major forms of liver injury including NAFLD, alcoholic-liver disease (ALD), and viral hepatitis-induced fibrosis (*Buch et al., 2015*; *Mancina et al., 2016*; *Luukkonen et al., 2016*; *International Liver Disease Genetics Consortium et al., 2016*; *Viitasalo et al., 2016*; *Krawczyk et al., 2017*; *Thabet et al., 2017*). The rs641738 variant is associated with a C > T missense single nucleotide polymorphism (SNP) within the first exon the *TMC4* gene, yet available data suggest that *TMC4* is not abundantly expressed in human liver (*Mancina et al., 2016*). Based on eQTL studies in the liver it has been suggested that instead reduced expression and activity of *MBOAT7* may be mechanistically linked to liver disease progression (*Mancina et al., 2016*; *Luukkonen et al., 2016*; *International Liver Disease Genetics Consortium et al., 2016*; *Viitasalo et al., 2016*; *Krawczyk et al., 2017*; *Thabet et al., 2017*).

The *MBOAT7* gene encodes an acyltransferase enzyme that specifically esterifies arachidonyl-CoA to lysophosphatidylinositol (LPI) to generate the predominant molecular species of phosphatidylinositol (PI) in cell membranes (38:4) (*Gijón et al., 2008*; *Zarini et al., 2014*; *Lee et al., 2012*; *Anderson et al., 2013*). Given this biochemical activity, MBOAT7 is a unique contributor to the Lands' cycle, which is a series of phospholipase-driven deacylation and lysophospholipid acyltransferase-driven reacylation reactions that synergize to alter phospholipid fatty acid composition, creating membrane asymmetry and diversity (*Shindou and Shimizu, 2009*; *Shindou et al., 2009*). It is important to note that MBOAT7, unlike other lysophospholipid acyltransferases, only diversifies the fatty acid composition of membrane PI species and not phospholipids with other head groups (*Gijón et al., 2008*; *Zarini et al., 2014*; *Lee et al., 2012*; *Anderson et al., 2013*). Despite recent progress in characterizing the selective biochemistry of MBOAT7 (*Gijón et al., 2008*), and the clear genetic links to liver disease (*Buch et al., 2015*; *Mancina et al., 2016*; *Luukkonen et al., 2016*; *International Liver Disease Genetics Consortium et al., 2016*; *Viitasalo et al., 2016*; *Krawczyk et al., 2017*; *Thabet et al., 2017*), there is no information regarding how MBOAT7 activity or its substrate (LPI) or product (PI) lipids impact liver disease progression. Here, we demonstrate that MBOAT7 expression is suppressed in obese humans and rodents. Furthermore, we show that *Mboat7*, but not *Tmc4*, loss of function in mice is sufficient to drive NAFLD progression, and show that *Mboat7* substrate lipids (LPIs) may be critical mediators of obesity-linked liver disease progression.

**eLife digest** Non-alcoholic fatty liver disease, or NAFLD for short, is a medical condition that develops when the liver accumulates excess fat. It can lead to complications such as diabetes and liver scarring. In humans, mutations that inactivate a protein called MBOAT7 increase the risk of fat accumulating in the liver.

Genetic studies suggest that low levels of MBOAT7 in a human's liver cells increase the severity of NAFLD. Yet the links between MBOAT7, NAFLD and obesity are not well understood. Helsley et al. used data from humans and from obese mice that had been fed a high-fat diet to investigate the relationship between NAFLD and MBOAT7. This revealed that people who are obese have lower levels of MBOAT7 in their livers. Next, obese mice were genetically manipulated to produce less MBOAT7, which led them to develop more severe NAFLD.

Helsley et al. then grew human liver cells in the laboratory and lowered their levels of MBOAT7, which led to excess fat accumulating in the cells. This increase in fat accumulation was, at least in part, due to how these cells metabolize fats when MBOAT7 is reduced: they start making more new fats and consume fewer lipids to produce energy.

These findings provide a link between obesity and liver damage in both humans and mice, and show how a decrease in MBOAT7 levels causes changes in fat metabolism that could lead to NAFLD. The results could drive new approaches to treating liver damage in patients with mutations in the gene that codes for MBOAT7.

DOI: https://doi.org/10.7554/eLife.49882.002

## Results

### *MBOAT7* expression is suppressed in obese humans and rodents

NAFLD progression is commonly associated with obesity, and it has even been suggested that obesity is a causative factor in the pathogenesis of NAFLD (*Younossi et al., 2018*). Although, there is now strong support that the common *MBOAT7* SNP (rs641738) is associated with NAFLD, it is not known whether *MBOAT7* expression is significantly altered in obese people. To investigate this, we obtained wedge liver biopsies from sequentially consenting obese bariatric surgery patients and normal weight controls (*Schugar et al., 2017*), and examined the expression levels of *MBOAT7* mRNA. We found that hepatic *MBOAT7* expression was dramatically reduced in obese people, when compared to normal weight controls (*Figure 1A*). It is important to note that this striking reduction in *MBOAT7* expression was not due to presence of the rs641738 *MBOAT7* SNP, as we found the minor allele frequency was quite similar in lean, obese, and severely obese subjects (*Figure 1A*). To follow up on our findings in human obesity, we also examined hepatic expression levels of *Mboat7* in obese leptin-deficient mice and high fat diet-fed rats. *Mboat7* expression was reduced in ob/ob mice compared to lean WT controls (*Figure 1B*). In agreement, we found that high fat diet-induced obesity in Sprague-Dawley rats was also associated with significant reductions in hepatic *Mboat7* expression (*Figure 1C*). Collectively, these results demonstrate that hepatic expression of *MBOAT7* is suppressed in obese humans and rodents.

Next, we used a systems genetics approach to examine links between *Mboat7* expression and adiposity traits in mice by leveraging data generated using the hybrid mouse diversity panel (HMDP) (*Ghazalpour et al., 2012*). To induce obesity, all HMDP mouse strains were fed an obesity-promoting high fat and high sucrose diet (*Parks et al., 2013*). Across the different strains in the HMDP, we found that *Mboat7* expression in the liver was only modestly correlated (r = −0.244, p=0.01) with adiposity (*Figure 1D*). However, *Mboat7* expression in white adipose tissue (WAT) was strongly negatively correlated (r = −0.68, p=2.3e-15) with both fat pad weight and % body fat in male and female mice (*Figure 1E*). Given the fact that obesity is commonly associated with insulin resistance and type 2 diabetes mellitus (*Cohen et al., 2011*; *Machado and Diehl, 2016*; *Rinella and Sanyal, 2016*; *Younossi et al., 2018*), we also examined the relationship between *Mboat7* expression and insulin sensitivity in the HMDP. We found that expression of *Mboat7* in adipose tissue was negatively (r = −0.38, p=0.0002) associated with indices of insulin sensitivity (*Figure 1F*). Collectively, these

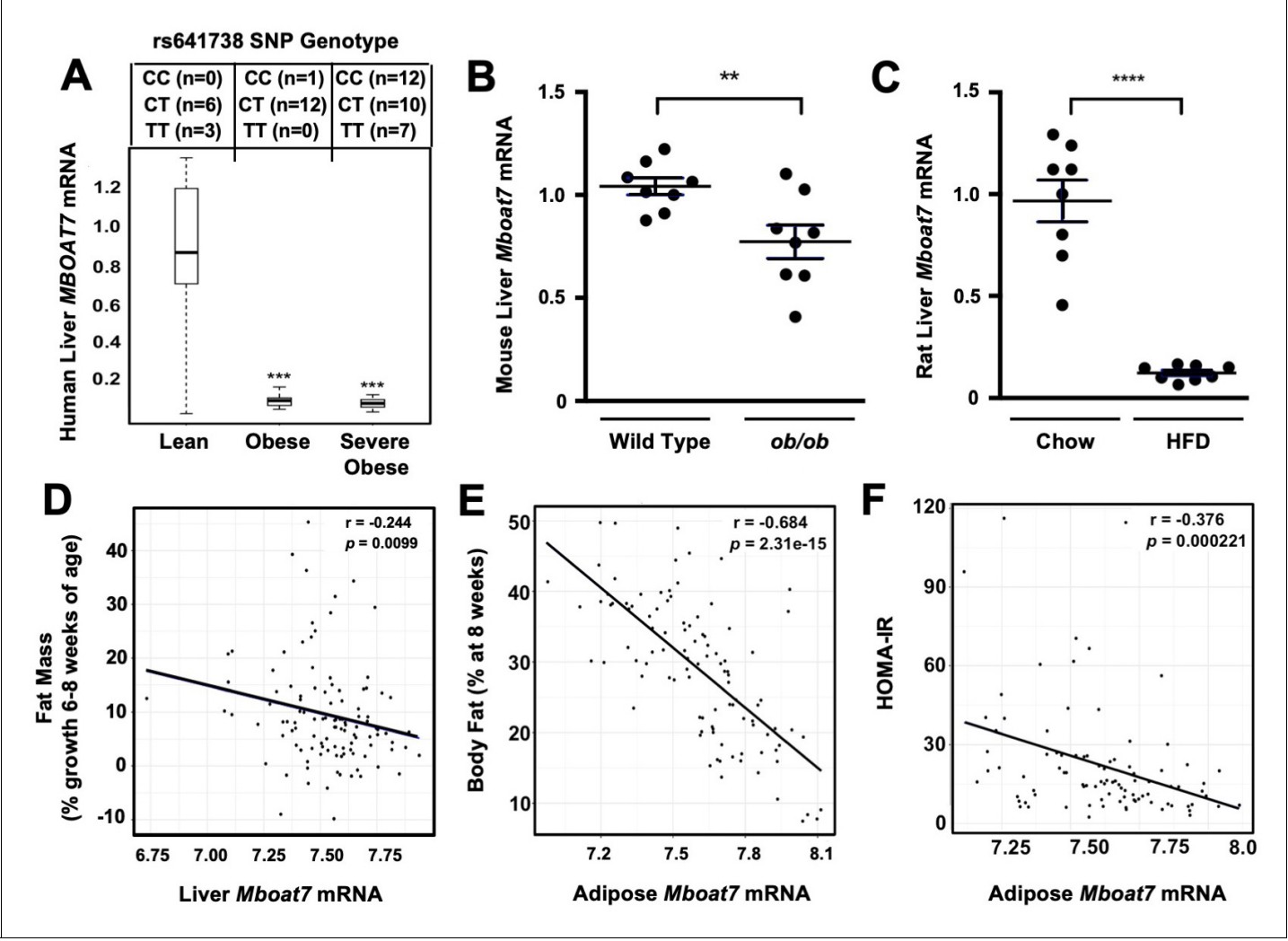

**Figure 1.** Obesity suppress MBOAT7 expression in humans, mice, and rats. (**A**) To investigate the impact human obesity has on MBOAT7 expression, we obtained wedged liver biopsies from sequentially consenting bariatric surgery patients and normal weight controls. We used these biopsies to isolate RNA and measure MBOAT7 expression via qPCR. The data are stratified based on BMI. A total of 52 liver biopsies were analyzed. Lean (BMI <30; n = 10), Obese (30 < BMI > 40; n = 13), and Severely Obese (BMI >40; n = 29) ***p≤0.001 compared to the lean group. Status of the rs641738 SNP is shown in the top inset. (**B**) Liver MBOAT7 expression was measured by QPCR from wild type and *ob/ob* mice 16–20 weeks of age (n = 8; **p≤0.01; two-tailed *t*-test; Data are presented as mean ± S.E.M.) (**C**) Liver MBOAT7 expression was measured by QPCR from Sprague Dawley rats fed a low fat or high fat high diet for 24 weeks (n = 8; **p≤0.01; two-tailed *t*-test; Data are presented as mean ± S.E.M.). (**D–F**) We used a systems genetics approach to examine links between MBOAT7 expression and metabolic traits in mice from the hybrid mouse diversity panel (HMDP). To induce obesity, all mouse strains represented in the HMDP were fed an obesity-promoting high fat and high sucrose diet. Across the different strains in the HMDP, we found that the expression of MBOAT7 in adipose tissue was strongly negatively correlated with body fat percentage (**E**), and HOMA-IR (**F**), while liver MBOAT7 expression was also negatively correlated with fat mass (**D**).

DOI: https://doi.org/10.7554/eLife.49882.003

data suggest that MBOAT7 may be mechanistically linked to the well-known association between obesity, insulin resistance, and NAFLD progression.

## *Mboat7* loss of function in mice is sufficient to promote NAFLD progression

To test whether MBOAT7 impacts obesity-linked NAFLD progression we utilized an in vivo knock-down approach in high fat diet-fed C57BL/6 mice. Metabolic phenotyping of global *Mboat7⁻/⁻* mice has been limited due to the fact that these mice are only viable for several weeks after birth due to the critical role *Mboat7* plays in cortical lamination and neuronal migration in the brain (***Lee et al.,***

*2012*; *Anderson et al., 2013*). To overcome this barrier, we have generated second-generation anti-sense oligonucleotides (ASOs), which predominately target liver, adipose tissue, and cells within the reticuloendothelial system to selectively knock down *Mboat7* expression in adult mice using methods we have previously described (*Schugar et al., 2017*). This ASO approach allows us to circumvent the postnatal lethality of global *Mboat7* deletion (*Lee et al., 2012*; *Anderson et al., 2013*), and permits investigation into obesity-linked liver disease progression with near complete loss of function of *Mboat7* in the liver. *Mboat7* ASO treatment resulted in significant reductions of *Mboat7* mRNA and protein in the liver (*Figure 2A*), and white adipose tissue (*Figure 2—figure supplement 1*), without altering *Mboat7* expression in the brain, spleen, heart, or skeletal muscle (*Figure 2—figure supplement 2*). Although *Mboat7* ASO treatment significantly reduced adipose *Mboat7* expression (*Figure 2*, *Figure 2—figure supplement 1*), this was not associated with alterations in body weight (*Figure 2B*), fat mass (*Figure 2—figure supplement 1L*), adipose gene expression (*Figure 2—figure supplement 1A–K*), food intake (*Figure 2C*), energy expenditure (*Figure 2D*, and *Figure 2—figure supplement 3*), or physical activity (*Figure 2—figure supplement 3*). Despite this lack of phenotypic differences in adipose tissue, *Mboat7* knockdown resulted in large alterations in the liver lipid storage. *Mboat7* ASO treatment promoted an increase in liver weight (*Figure 2F*) and striking hepatic steatosis in HFD-fed, but not chow-fed mice (*Figure 2E,I*). Importantly, *Mboat7* ASO-driven hepatic steatosis was consistently seen with two distinct ASOs targeting different regions of the *Mboat7* messenger RNA (*Figure 2—figure supplement 4*). *Mboat7* ASO-driven hepatic steatosis was characterized by accumulation of triglycerides, free cholesterol, and cholesterol esters only in high fat fed cohorts (*Figure 2J–M*). Despite these significant alterations in hepatic lipids, *Mboat7* knockdown did not dramatically alter the levels of triglycerides or cholesterol in circulating lipoproteins (*Figure 2—figure supplement 5*). *Mboat7* knockdown was also associated with hepatocyte injury as indicated by elevated liver enzyme levels (AST and ALT), but this only occurred in high fat-fed mice (*Figure 2G,H*).

To more comprehensively understand the global effects of *Mboat7* knockdown on liver function, we performed unbiased RNA sequencing experiments in control versus *Mboat7* ASO-treated mice (*Figure 3*). Principal component analysis of RNA expression profiles showed separation by ASO group according to principal component analysis and hierarchical clustering analyses (*Figure 3A,B*). ASO groups were also partitioned by unsupervised hierarchical clustering (data not shown). *Mboat7* knockdown resulted in a number of differentially expressed genes (DEGs), with 124 DEGs being suppressed and 887 DEGs being upregulated by *Mboat7* ASO treatment (top 50 DEGs shown in *Figure 3B* and overall changes in *Figure 3C*). To understand the major pathways affected by *Mboat7* knockdown, we performed Gene Ontology Molecular Function enrichment analysis, and found that many of the pathways that were significantly enriched are mechanistically linked to liver injury such as leukocyte extravasation, monocyte/macrophage activation, and fibrosis/hepatic stellate cell activation (*Figure 3D*). As predicted *Mboat7* mRNA levels were dramatically suppressed by *Mboat7* ASO treatment, but quite unexpectedly we also saw that the expression of *Tmc4* was reduced in *Mboat7* ASO-treated mice (*Figure 3B,E*). These data shown for the first time that ASO-mediated knockdown of *Mboat7* results in coordinated suppression of its neighboring gene *Tmc4* (the rs641738 polymorphism is located in exon 1 of the *TMC4* gene), indicating some potential cross talk that deserves further exploration (*Figure 3E*). Quantitative real time PCR (qPCR) analysis also confirmed the RNA sequencing results showing that *Mboat7* knockdown resulted in elevated expression of gene associated with inflammation (*Tnfa* and *Il1b*) and early fibrosis (*Col1a2* and *Acta2*) in high fat fed mice (*Figure 3E*). In addition, *Mboat7* knockdown was associated with altered immune cell populations in the liver. *Mboat7* knockdown increased Cd8+ T lymphocytes, while decreasing the total number of Cd11b+ macrophages in the liver (*Figure 3—figure supplement 1*). Of the hepatic macrophages that were present, more were skewed towards the proinflammatory M1 (Cd11c+) state, and less were skewed towards the alternative M2 (Cd206+) phenotype (*Figure 3—figure supplement 1*). Collectively, these data suggest that specifically under high fat feeding conditions, *Mboat7* loss of function is associated with dysregulated immune cell homeostasis, inflammation, and gene expression signatures that are consistent with early activation of pro-fibrotic programs in the liver (*Figure 3*).

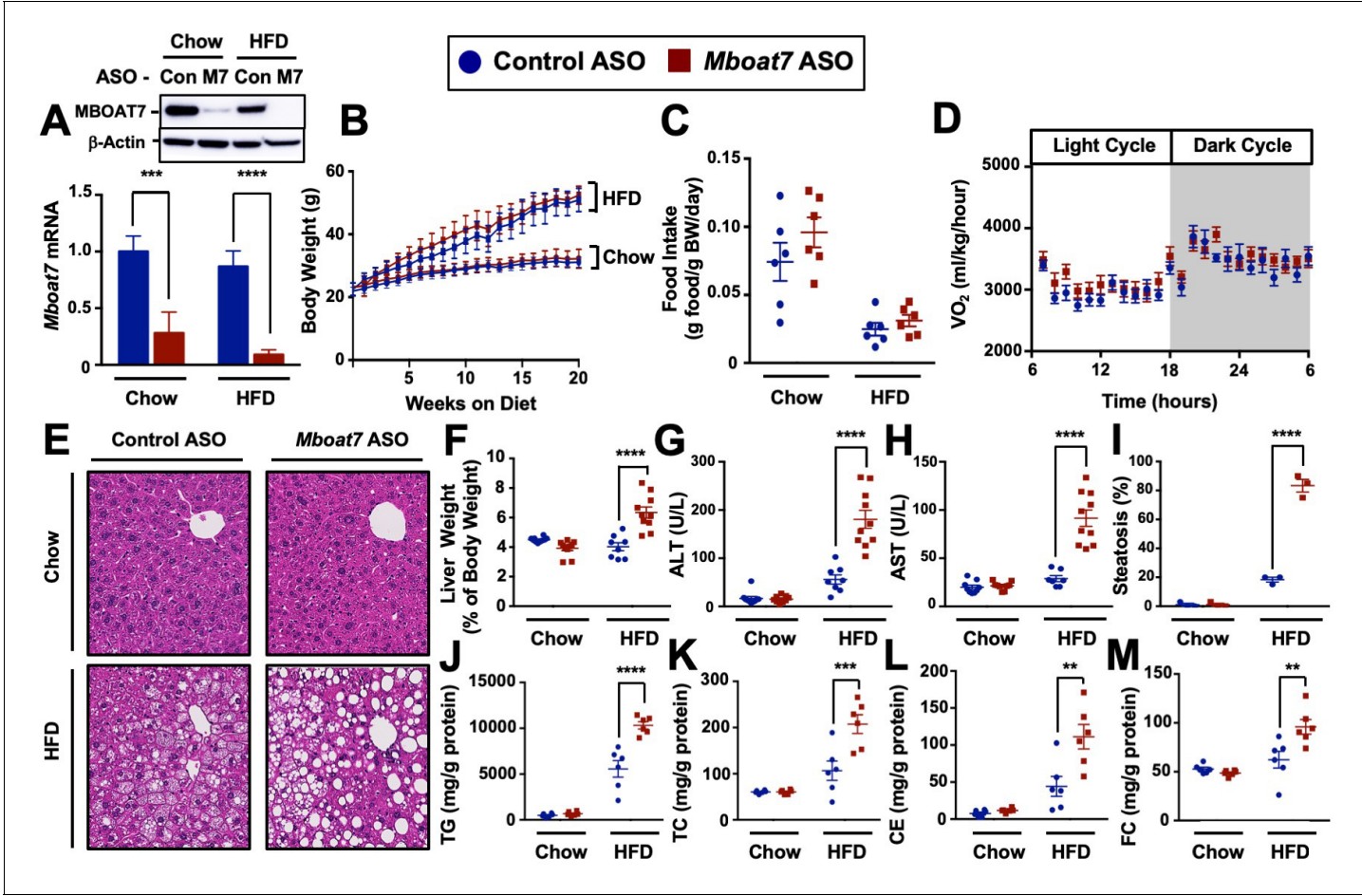

**Figure 2.** *Mboat7* knockdown promotes hepatic steatosis and worsen liver injury. (**A and B**) C57BL/6 mice were fed chow or HFD with concurrent Control and *Mboat7* ASO Injections at 12.5 mg/kg week for 20 weeks. Liver *Mboat7* expression was measured via qPCR (**A**); n = 8–10; *p≤0.05, ****p≤0.0001; Two-way ANOVA with Tukey's *post-hoc* test). The inset show a representative Western blot for hepatic MBOAT7 protein levels, which was replicated in n = 4 mice. (**B**) Body weight was measured weekly. (**C**) Food intake was measured in C57BL/6 mice at 12 weeks of diet and ASO injections (n = 6). (**D**) Oxygen consumption was measured by indirect calorimetry in C57BL/6 mice after 12 weeks of diet and ASO injections (n = 6). (**E**) Representative liver hematoxylin and eosin stained sections. 20x magnification. (**F**) Liver-to-body weight measurements from mice fed Chow and HFD with Control and *Mboat7* ASO Injections for 20 weeks (n = 8–10; ****p≤0.0001; Two-way ANOVA with Tukey's *post-hoc* test). (**G and H**) Plasma ALT (**G**) and AST (**H**) levels were measured after 20 weeks of diet feeding and ASO injections (n = 8–10; ****p≤0.0001; Two-way ANOVA with Tukey's *post-hoc* test). (**I**) Percent steatosis quantified by a blinded board certified pathologist at the Cleveland Clinic (n = 3; ****p≤0.0001; Two-way ANOVA with Tukey's *post-hoc* test). (**J–M**) Hepatic triglycerides (**J**), hepatic cholesterol (**K**), hepatic esterified cholesterol (**L**), and hepatic free cholesterol (**M**) were measured colorimetrically (n = 8–10; **p≤0.01, ***p≤0.001, ****p≤0.0001; Two-way ANOVA with Tukey's *post-hoc* test). All data are presented as mean ± S.E.M., unless otherwise noted.

DOI: https://doi.org/10.7554/eLife.49882.004

The following figure supplements are available for figure 2:

**Figure supplement 1.** *Mboat7* ASO treatment reduces *Mboat7* expression in white adipose tissue (WAT), but does not dramatically alter WAT gene expression or adiposity.

DOI: https://doi.org/10.7554/eLife.49882.005

**Figure supplement 2.** Mboat7 ASO treatment does not alter Mboat7 expression in several extrahepatic tissues.

DOI: https://doi.org/10.7554/eLife.49882.006

**Figure supplement 3.** A reduction in hepatic MBOAT7 expression does not alter metabolic parameters.

DOI: https://doi.org/10.7554/eLife.49882.007

**Figure supplement 4.** Hepatic steatosis is observed in two distinct ASOs targeting different regions of the *Mboat7* mRNA.

DOI: https://doi.org/10.7554/eLife.49882.008

**Figure supplement 5.** Knockdown of *Mboat7* alters plasma lipid levels in chow but not HFD-fed mice.

DOI: https://doi.org/10.7554/eLife.49882.009

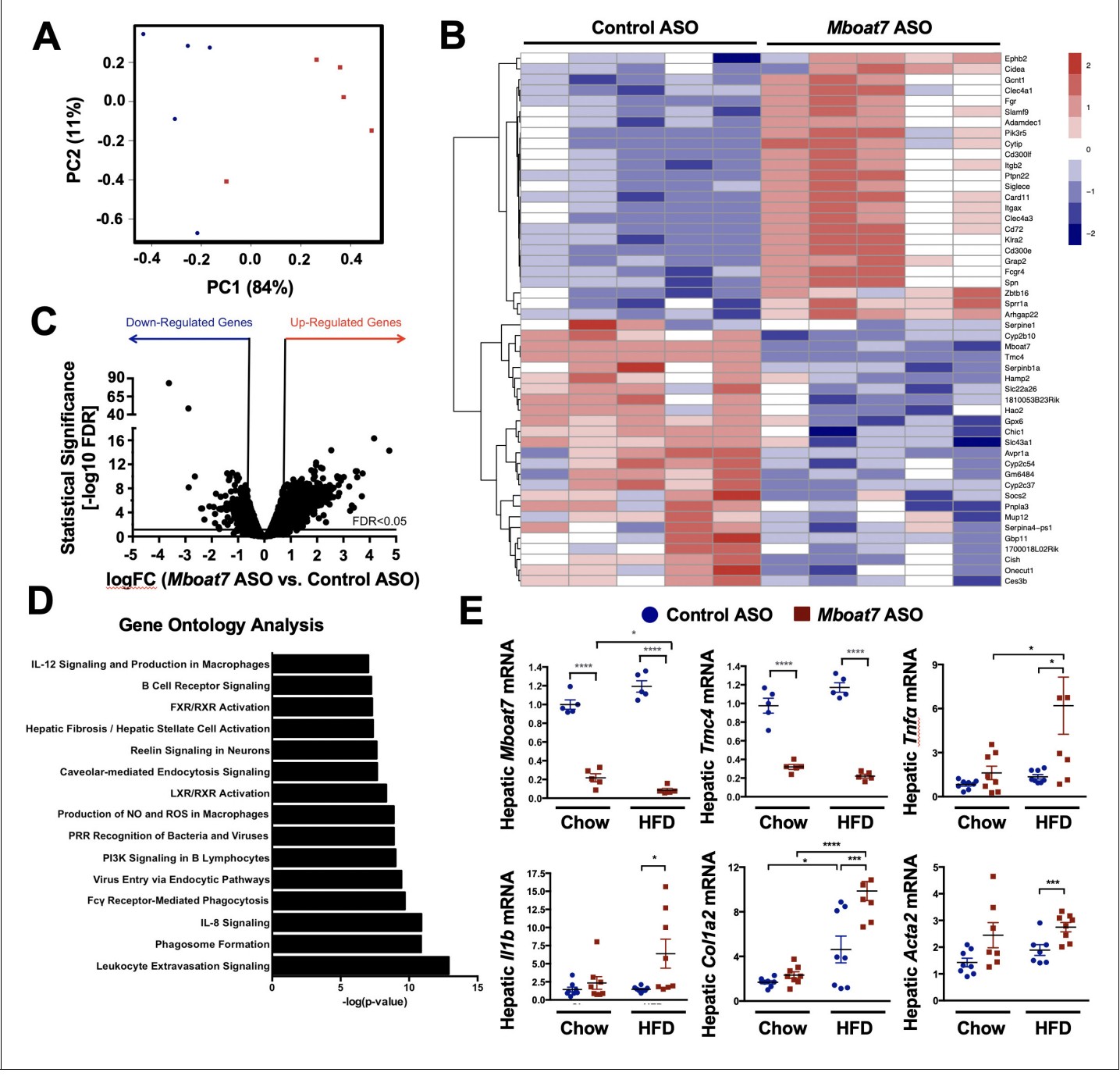

**Figure 3.** *Mboat7* inhibition exacerbates inflammation and fibrotic gene expression in the liver. (A–C) C57BL/6 mice were fed a HFD with concurrent Control and *Mboat7* ASO Injections at 12.5 mg/kg week for 20 weeks. Liver RNA was used for RNA-sequencing. ASO groups clustered by principal component analysis (A). Row-normalized expression for the top 50 DEGs are shown by heat map (B) while the volcano plot (C) summarizes log2 fold changes vs significance in response to *Mboat7* inhibition (n = 5; genes with FDR < 0.05 and fold change >2 were considered significantly differentially expressed). (D) Summary of significantly differentially regulated pathways in mice treated with a non-targeting Control ASO vs the *Mboat7* ASO. (E) qPCR validation of selected gene expression changes discovered in the RNA-sequencing analysis in both Chow and HFD-fed mice receiving Control or MBOAT7 ASOs for 20 weeks (n = 8–10; *p≤0.05, ***p≤0.001, ****p≤0.0001; Two-way ANOVA with Tukey's *post-hoc* test). All data are presented as mean ± S.E.M., unless otherwise noted.

DOI: https://doi.org/10.7554/eLife.49882.010

The following figure supplement is available for figure 3:

**Figure supplement 1.** A reduction in *Mboat7* expression alters hepatic immune cell populations.

*Figure 3 continued*

DOI: https://doi.org/10.7554/eLife.49882.011

## Genetic deletion of Tmc4 does not result in hepatic steatosis

Given the fact that the rs641738 polymorphism is located in exon 1 of the *TMC4* gene, and we unexpectedly found that *Mboat7* ASO treatment also reduces *Tmc4* expression (*Figure 3B,E*), we wanted to examine whether alteration in *Tmc4* may also be a key regulator of hepatic steatosis. To understand the role of *Tmc4* in hepatic steatosis, we generated global *Tmc4* knockout mice using CRISPR-Cas9-mediated gene editing and examined hepatic lipid levels under high fat feeding conditions (*Figure 4*). Global *Tmc4* knockout mice have marked reductions in hepatic *Tmc4* mRNA (*Figure 4A*) and protein (*Figure 4B*), yet importantly *Mboat7* expression is unaltered (*Figure 4A*). In contrast to the striking hepatic steatosis seen with *Mboat7* loss of function (*Figure 2E*, *Figure 2—figure supplement 4*), *Tmc4* null mice show similar levels of hepatic lipids when fed a high fat diet (*Figure 4*). These data strongly implicate *Mboat7*, but not *Tmc4*, as the primary mediator of hepatic steatosis seen with the rs641738 variant or in *Mboat7* ASO-treated mice.

## *Mboat7* knockdown promotes profound hyperinsulinemia and impairs hepatic insulin action

The abnormal accumulation of lipids (i.e., lipotoxicity) in tissues such as the pancreas and liver is frequently associated with the pathogenesis of type 2 diabetes (*Unger and Scherer, 2010*;

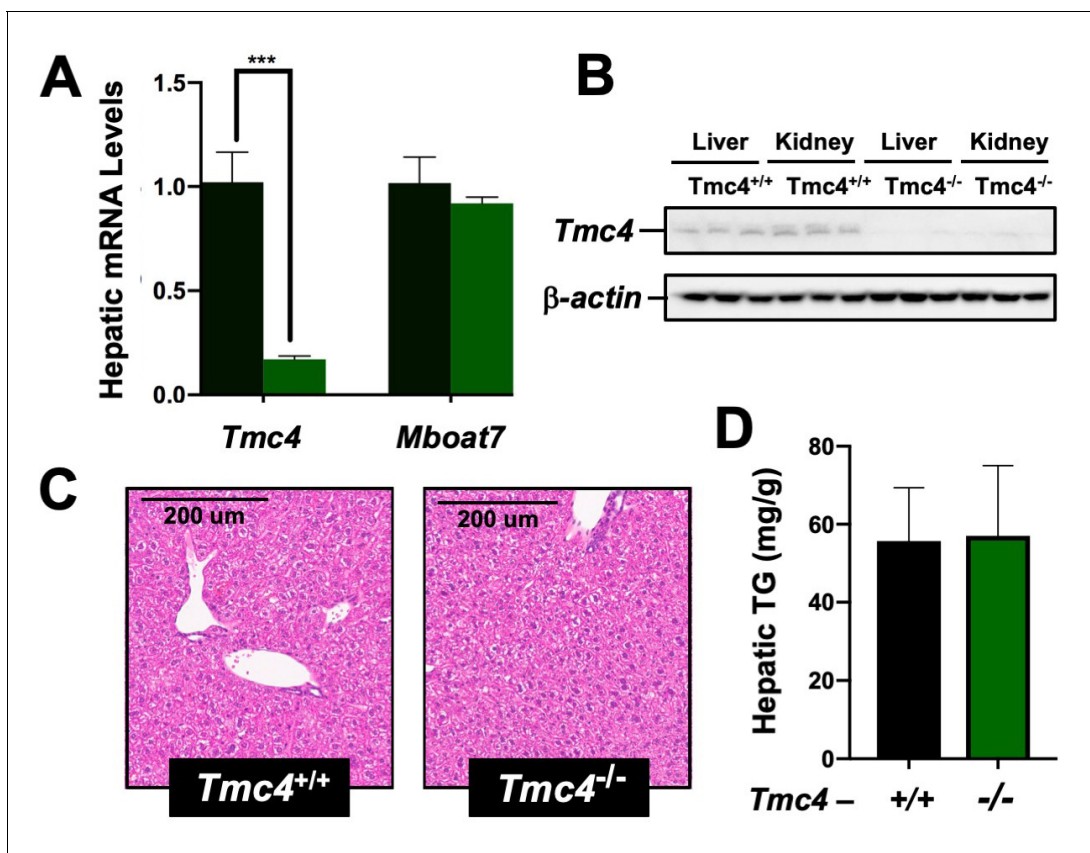

**Figure 4.** *Tmc4* knockout mice do not exhibit fatty liver. Male wild type (*Tmc4*[+/+]) or *Tmc4* global knockout mice (*Tmc4*[-/-]) were fed a high fat diet for 2 weeks. (A) Hepatic gene expression for *Tmc4* and *Mboat7* measured by qPCR. (B) Western blot of TMC4 protein levels in the liver and kidney. (C) Representative H and E-stained liver sections from *Tmc4*[+/+] and *Tmc4*[-/-]. (D) Hepatic triglyceride (TG) levels in *Tmc4*[+/+] and *Tmc4*[-/-] were quantified biochemically. Data shown represent n = 5–6 mice per group, ***p≤0.001; two-tailed *t*-test relative to Tmc4[+/+] mice.

DOI: https://doi.org/10.7554/eLife.49882.012

*Samuel and Shulman, 2018*). Given that *Mboat7* knockdown promoted striking hepatic steatosis (*Figure 2E*), and the fact that adipose tissue expression of *Mboat7* is negatively correlated with insulin sensitivity across strains of the HMDP (*Figure 1F*) we examined glucose homeostasis in *Mboat7* ASO-treated mice fed a high fat diet (*Figure 5*). *Mboat7* knockdown was associated with significantly impaired systemic glucose tolerance (*Figure 5A*). *Mboat7* ASO-treated mice also exhibited profound hyperinsulinemia in the fasted state and throughout an intraperitoneal glucose tolerance test (*Figure 5B*). In parallel, glucose-stimulated C-peptide release was elevated in *Mboat7* ASO-treated mice, potentially indicating overproduction of insulin in the pancreatic β-cell (*Figure 5C*). During an intraperitoneal insulin tolerance test, *Mboat7* ASO-treated mice showed blunted plasma glucose lowering (*Figure 5D*). To more directly measure tissue-specific insulin action in *Mboat7* knockdown mice we examined the acute phosphorylation of the insulin receptor (IR) and protein kinase B (Akt) in response to a single insulin injection (*Figure 5E–G*). *Mboat7* ASO-treated mice had reduced insulin-stimulated phosphorylation of IR and downstream Akt in the liver (*Figure 5E*). However, *Mboat7* ASO treatment did not significantly alter insulin signal transduction in either skeletal muscle or white adipose tissue (*Figure 5F,G*). These data suggest that ASO-mediated knockdown of *Mboat7* promotes hyperinsulinemia, yet impairs insulin action specifically in the liver.

## ASO-mediated knockdown of *Mboat7* results in tissue-specific alterations in LPI and PI lipids

Given the fact that *Mboat7* is known to catalyze the selective esterification of arachidonyl-CoA to LPI lipids in neutrophils (*Gijón et al., 2008*), we wanted to examine its lysophosphatidylinositol acyltransferase (LPIAT) activity in the liver and also examine both substrate LPI and product PI species across a range of tissues. First, we isolated hepatic microsomes from control and *Mboat7* ASO-treated mice and assayed LPIAT enzymatic activity using saturated and monounsaturated LPI substrates (*Figure 6A*). In chow fed mice *Mboat7* knockdown only modestly decreased hepatic LPIAT activity toward 18:1 LPI (*Figure 6A*), yet in high fat diet-challenged mice *Mboat7* knockdown resulted in a highly significant ~50% reduction in LPIAT activity using either 16:0, 18:0, or 18:1 LPI substrates (*Figure 6A*). These data suggest that *Mboat7* is a significant contributor to total hepatic LPIAT activity, especially under high fat feeding conditions. Aligned with these alterations in enzyme activity, knockdown of *Mboat7* in high fat-fed mice resulted in alterations in LPI and PI lipids in a highly tissue-specific manner. *Mboat7* ASO treatment did not significantly alter LPI levels in the circulation, but was associated with selective reduction in the 38:3 and 38:4 species of circulating PI lipids (*Figure 6B*, *Figure 6C*, and *Figure 6—figure supplements 1–2*). In contrast to effects in the circulation, *Mboat7* knockdown was associated with significant accumulation of 16:0 and 18:1 LPI species in the liver (*Figure 6D* and *Figure 6—figure supplement 2*). Furthermore, *Mboat7* ASO-treatment resulted in reduced 38:3 and 38:4 PI levels, while increasing more saturated (34:1, 34:2, 36:1, and 36:2) PI species in the liver (*Figure 6E* and *Figure 6—figure supplement 1*). In white adipose tissue, *Mboat7* knockdown did not significantly alter LPI levels (*Figure 6—figure supplement 2*), but promoted marked accumulation of more saturated PI species both in chow and high fat fed mice (*Figure 6—figure supplement 1*). Also, *Mboat7* ASO treatment resulted in significant reductions in several PUFA-enriched PI species (38:3, 38:4, 38:5, and 38:6) in adipose tissue (*Figure 6—figure supplement 1*). In contrast to alterations in LPI and PI levels in liver and adipose tissue (*Figure 6* and *Figure 6—figure supplements 1–2*), *Mboat7* ASO treatment did not significantly alter LPI or PI levels in the brain (*Figure 6—figure supplement 3*) or pancreas (*Figure 6—figure supplement 4*). In contrast to marked changes in inositol-containing phospholipids, it is also important to note that *Mboat7* knockdown did not significantly alter the hepatic levels of other major phospholipids including phosphatidylcholines, lysophosphatidylcholines, phosphatidylethanolamines, lysophosphatidylethanolamines, phosphatidylserines, phosphatidylglycerols, or phosphatidic acids (*Figure 6—figure supplement 5*). Collectively, these data show that ASO-mediated knockdown of *Mboat7* primarily alters LPI and PI levels in the liver and white adipose tissue, which creates the potential to induce an imbalance of local lipid mediators that originate from PI metabolism.

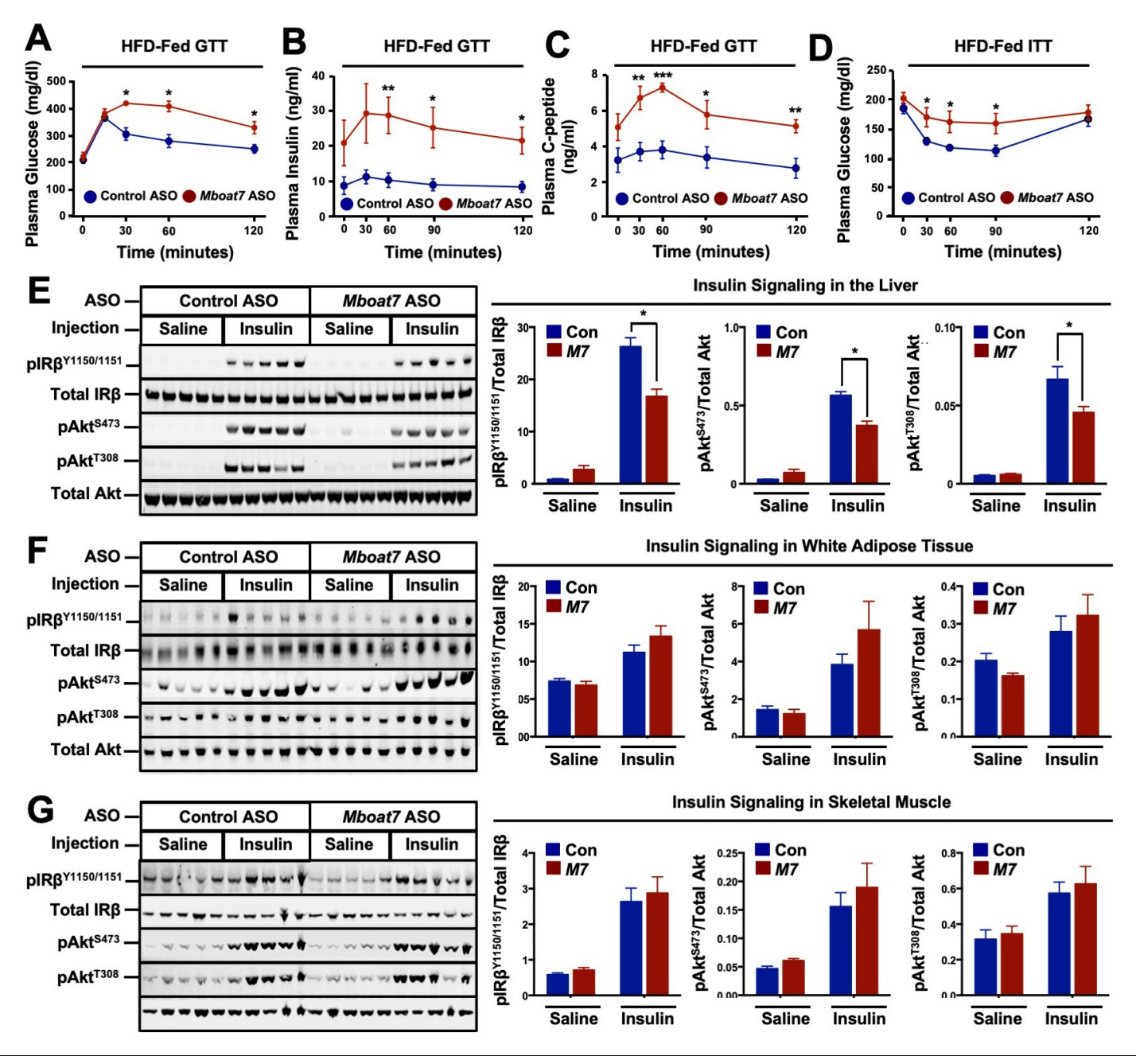

**Figure 5.** A reduction in *Mboat7* expression exacerbates hepatic insulin resistance in HFD-fed mice. (**A–C**) C57BL/6 mice were fed a HFD with concurrent Control and *Mboat7* ASO Injections at 12.5 mg/kg week for 20 weeks. Plasma glucose (**A**), plasma insulin (**B**), and plasma C-peptide (**C**) levels were measured throughout the course of an intraperitoneal glucose tolerance test (n = 4–5; *p≤0.05, **p≤0.01; two-tailed *t*-test relative to control ASO at same time point). (**D**) Plasma glucose was measured after an intraperitoneal insulin injection in mice fed HFD for 20 weeks with Control and *Mboat7* ASO injections (n = 4–5; *p≤0.05, **p≤0.01; two-tailed *t*-test relative to control ASO at same time point). (**E–G**) Mice were fed a HFD for 20 weeks while injected with Control or *Mboat7* ASOs, then saline or insulin (0.35 U/kg) was injected into the portal vein for 5 min. The insulin signaling pathway was examined via western blot in tissues: liver (**E**), skeletal muscle (**F**), and white adipose tissue (**G**) (n = 5) each sample represents an independent mouse; western blot is quantified by densitometry on the right; *p≤0.05; Two-way ANOVA with Tukey's *post-hoc* test). All data are presented as mean ± S.E.M., unless otherwise noted.

DOI: https://doi.org/10.7554/eLife.49882.013

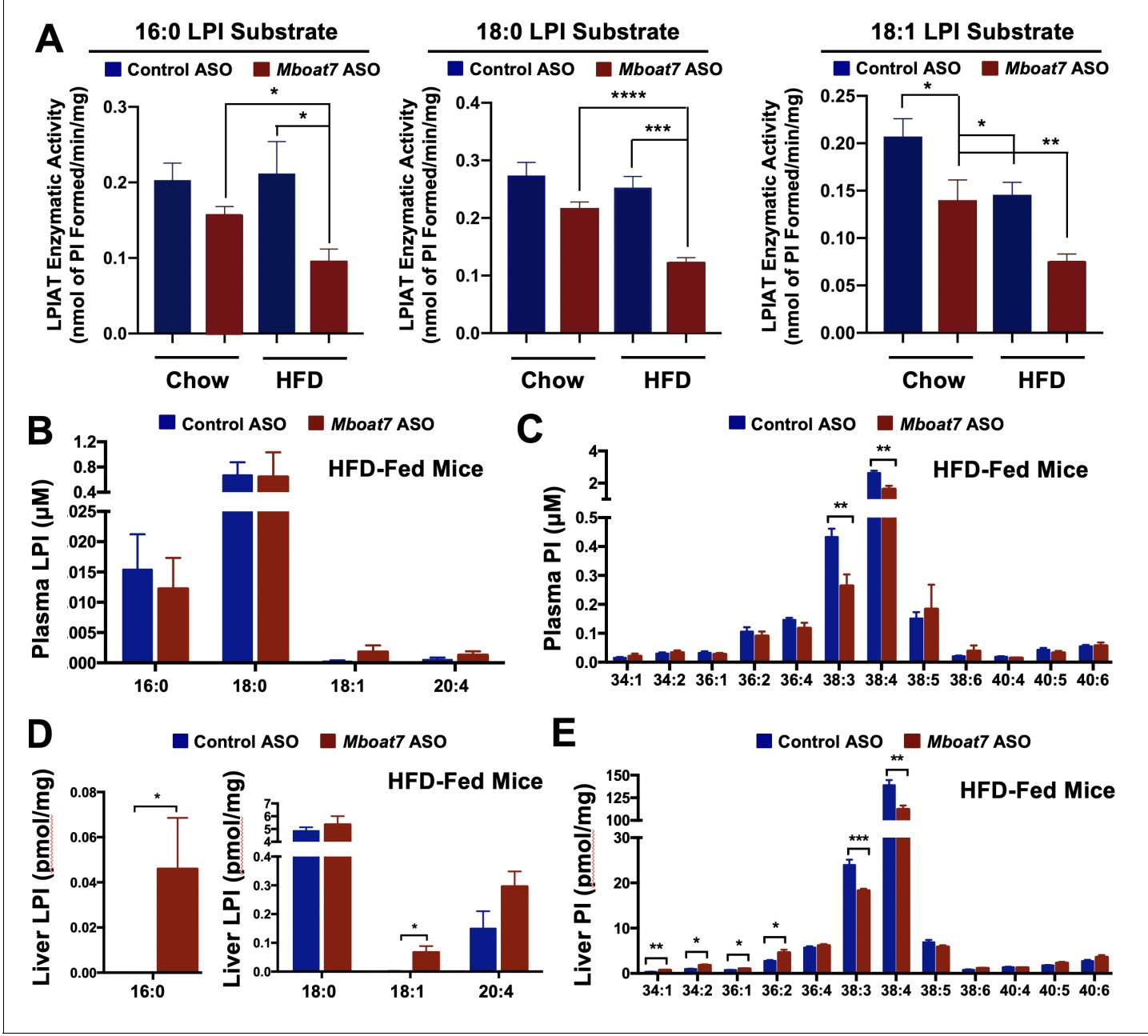

**Figure 6.** HFD-feeding and *Mboat7* inhibition results in an accumulation of LysoPIs. Male C57BL/6 mice were fed either chow or a HFD for 20 weeks along with receiving a control non-targeting ASO or an ASO targeting the knockdown of *Mboat7*. (**A**) Liver microsomes were isolated and assayed for lyso-phosphatidylinositol acyltransferase (LPIAT) activity, using [14C]-arachidonyl-CoA and either 16:0 LPI, 18:0 LPI, or 18:1 LPI as substrates. Plasma LPI (**B**) or PI (**C**) species were measured by LC-MS/MS (n = 5; *p≤0.05; two-tailed *t*-test). Liver LPI (**D**) and PI (**E**) species were measured by LC-MS/MS (n = 5–10 per group; *p≤0.05, **p≤0.01, ***p≤0.001; two-tailed *t*-test). All data are presented as mean ± S.E.M.

DOI: https://doi.org/10.7554/eLife.49882.014

The following figure supplements are available for figure 6:

**Figure supplement 1.** *Mboat7* ASO treatment impacts PI metabolism.
DOI: https://doi.org/10.7554/eLife.49882.015
**Figure supplement 2.** *Mboat7* ASO treatment impacts LPI metabolism.
DOI: https://doi.org/10.7554/eLife.49882.016
**Figure supplement 3.** *Mboat7* ASO treatment does not influence brain LPI and PI levels.
DOI: https://doi.org/10.7554/eLife.49882.017
**Figure supplement 4.** *Mboat7* ASO treatment does not influence pancreas LPI and PI levels.
*Figure 6 continued on next page*

*Figure 6 continued*

DOI: https://doi.org/10.7554/eLife.49882.018

**Figure supplement 5.** *Mboat7* ASO treatment does not significantly alter other major phospholipid species.
DOI: https://doi.org/10.7554/eLife.49882.019

**Figure supplement 6.** *Mboat7* ASO treatment does not drastically influence AA-derived lipid mediators.
DOI: https://doi.org/10.7554/eLife.49882.020

## LPI lipids increase hepatic inflammatory and fibrotic gene expression in an *Mboat7*-dependent manner

The major enzymatic product of MBOAT7 (38:4 PI) is a potential reservoir for the generation of arachidonic acid-derived immunomodulatory lipid mediators (*Serhan et al., 2015*; *Dennis and Norris, 2015*). Therefore, we initially hypothesized that *Mboat7* loss of function might limit the pool of arachidonic acid in PI lipids available for arachidonic acid-derived lipid mediator production. However, when we examined arachidonic acid-derived pro-inflammatory (LTB$_4$, PGE$_2$, PGD$_2$, PGF$_{2\alpha}$, and TXB$_2$) and pro-resolving (15$R$-LXA$_4$) lipid mediators there was no apparent alteration in the liver of *Mboat7* ASO-treated mice (*Figure 6—figure supplement 6*). Collectively, these data do not support a rate-limiting role for MBOAT7 in arachidonic acid-derived lipid mediator production in mice. Given the fact that *Mboat7* knockdown is associated with increases in LPI lipids, we next hypothesized that MBOAT7 substrate LPI lipids themselves may be the main drivers of liver disease progression under conditions of HFD-induced obesity and *Mboat7* loss of function. In further support of this concept, we found that saturated LPI lipids are significantly elevated in the circulation of humans with advanced fibrosis compared to healthy controls (*Figure 7*).

To more directly test the hypothesis that LPI lipids may promote liver disease in a *Mboat7*-dependent manner, we directly treated mice with LPI lipids to transiently increase circulating LPI levels

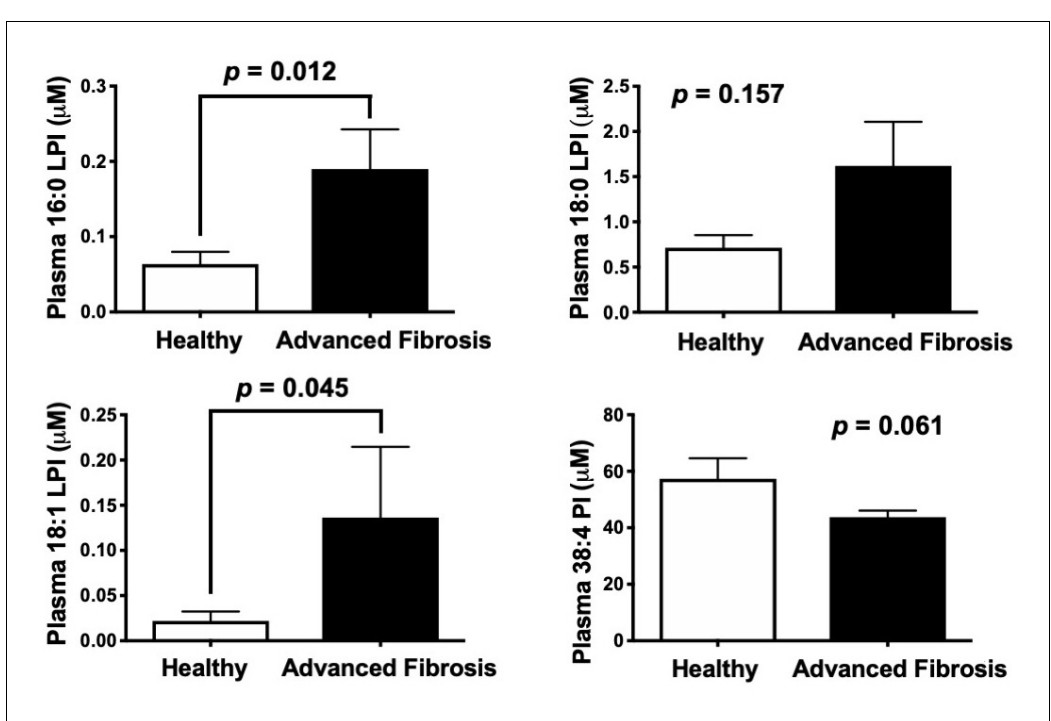

**Figure 7.** Circulating levels of saturated lysophosphatidylinositol (LPI) lipids are increased in humans with advanced fibrosis. Plasma from healthy subjects with no fibrosis (Ishak score = 0) or pathology proven advanced fibrosis (Ishak score = 4) were analyzed by LC-MS/MS to quantify MBOAT7 substrates (16:0 LPI, 18:0 LPI, and 18:1 LPI) and product (38:4 PI) lipids. All data are presented as mean ± S.E.M from n = 10 subjects per group, and group differences were determined using Mann-Whitney testing.
DOI: https://doi.org/10.7554/eLife.49882.021

within a physiologic level (~2 fold) (*Figure 8—figure supplement 1*). Injection of 16:0, 18:0, or 18:1

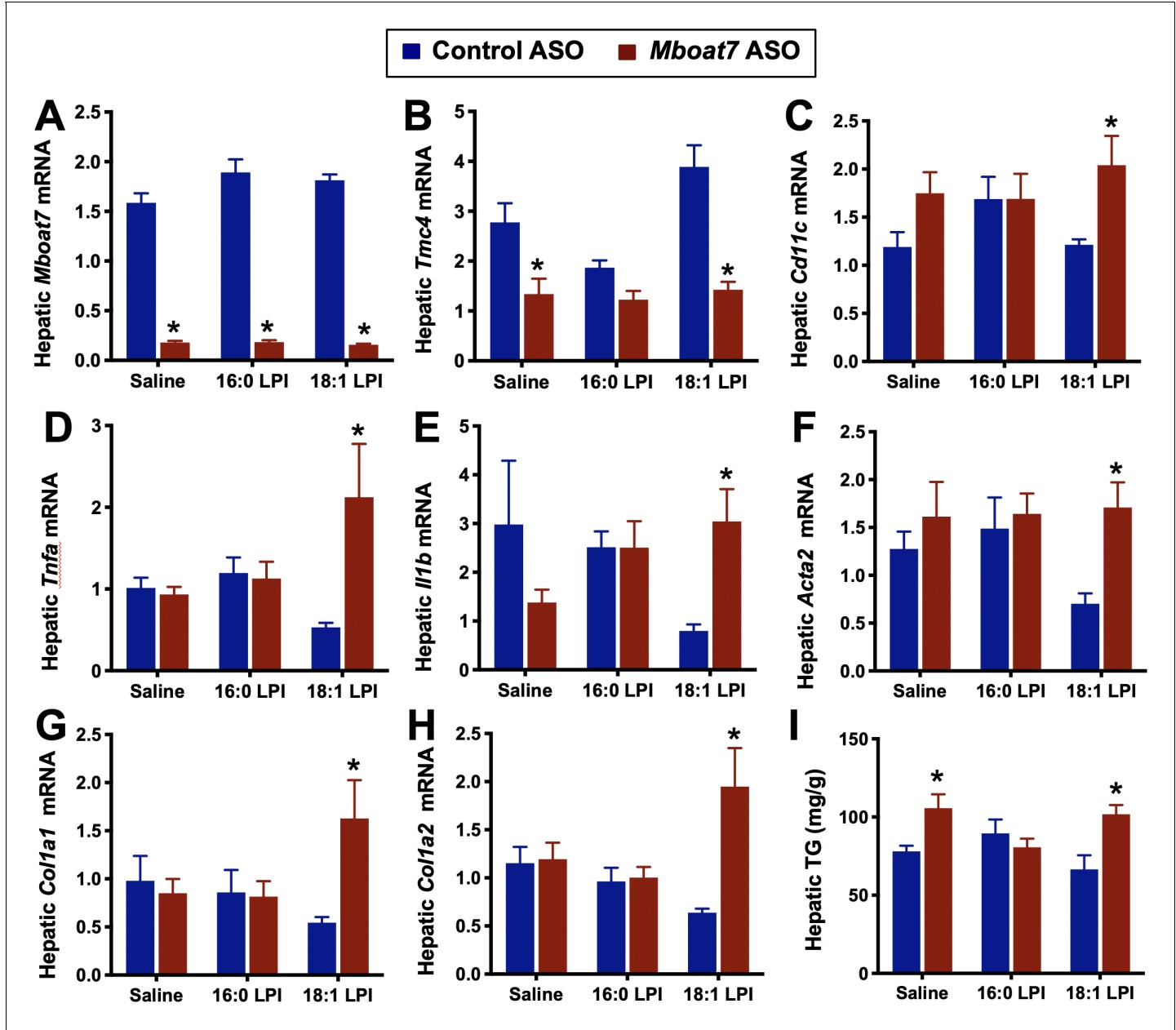

**Figure 8.** Lysophosphatidylinositol (LPI) lipids stimulate pro-inflammatory and pro-fibrotic gene expression in a *Mboat7*-dependent manner. Male C57BL/6 mice were treated with either a non-targeting control ASO or *Mboat7* ASO while being fed a high fat diet for 11 weeks. Thereafter, mice were injected IP with saline (vehicle) or 12.5 µg of the *Mboat7* substrate lipids 16:0 or 18:1 LPI (2 injections at 7 am and seven pm), and liver was collected 12 hr later. (A–H) Liver mRNA levels were measured by qPCR. (I) Liver triacylglycerol (TG) levels. Data are presented as mean ± S.E.M. (n = 5; *p≤0.05).

DOI: https://doi.org/10.7554/eLife.49882.022

The following figure supplements are available for figure 8:

**Figure supplement 1.** Accumulation of LPI species elevate after 4 hr post-injection.
DOI: https://doi.org/10.7554/eLife.49882.023
**Figure supplement 2.** 18:0 LPI stimulates pro-inflammatory and pro-fibrotic gene expression in a Mboat7-dependent manner.
DOI: https://doi.org/10.7554/eLife.49882.024
**Figure supplement 3.** LPI lipids alter hepatic gene expression.
DOI: https://doi.org/10.7554/eLife.49882.025

LPI into control ASO-treated mice did not alter hepatic inflammatory or fibrotic gene expression (*Figure 8* and *Figure 8—figure supplement 2*). This is not surprising, given that it was previously reported that exogenously provided LPI is rapidly esterified into membrane PI pools (*Darnell and Saltiel, 1991*; *Darnell et al., 1991*; *Jackson and Parton, 2004*). In other words, under normal conditions MBOAT7 activity rapidly acylates LPI to divert this signaling lipid into a membrane PI storage pool. We next postulated that under conditions such as obesity where *Mboat7* activity is diminished, LPI-driven signaling can be sustained and promote liver injury. In support of this concept, direct injection of exogenous 18 carbon LPIs can rapidly (24 hr) increase the expression of genes characteristic of hepatic inflammation (*Cd11c, Tnfa, IL1b*) and early fibrosis (*Desmin, Col1A1, Col1A2*, and *Acta2*) in mice with *Mboat7* knockdown (*Figure 8* and *Figure 8—figure supplement 2*). These data suggest that 18 carbon LPI lipids can acutely induce hepatic inflammatory and fibrotic gene expression programs, but only when MBOAT7 function is compromised, as is seen in obesity or with loss of function variants like rs641738. It is important to note that 18 carbon LPI lipids can only significantly elicit such pro-inflammatory and pro-fibrotic effects in high fat fed mice (*Figure 8* and *Figure 8—figure supplement 2*), as we did not see this same effect in chow fed cohorts (not shown). Although direct administration of 18 carbon LPI lipids in chow fed mice did not significantly alter the same pro-inflammatory or pro-fibrotic genes as we found with *Mboat7* knockdown (*Figure 3*), we did find that a small subset of acute phase response genes (serum amyloid A genes, *Saa1* and *Saa2*) and other immunomodulatory genes (Gdf15, Ly6d, Lcn2, Socs2, etc.) were altered by 18:0 LPI treatment in a *Mboat7*-dependent manner (*Figure 8—figure supplement 3*). Interestingly, when liver triacylglycerol levels were examined in this LPI injection experiment, we only found significantly elevation in the saline and 18:1 LPI injection groups, but not in the 16:0 LPI injection group (*Figure 8I*). Collectively, these results suggest that 18 carbon LPI lipids can alter hepatic inflammatory transcriptional programs, but this is highly reliant on both diminished *Mboat7* expression/activity and high fat diet feeding.

### Steatosis Seen with *Mboat7* Knockdown is Associated with Reorganization of the Hepatic Lipid Droplet Lipidome and Proteome in vivo, and Alterations In Lipogenesis and Fatty Acid Oxidation in Cultured Cells

*Mboat7* loss of function results in a striking accumulation of neutral lipids including triglycerides and cholesteryl esters (*Figure 2E–I*), yet the mechanism(s) behind this are poorly understood. We therefore examined several potential mechanisms driving the mixed hepatic steatosis in *Mboat7* ASO-treated mice. First, we examined the expression of genes involved in lipogenesis, fatty acid oxidation, and cholesterol sensing and export under both fed and fasted conditions (*Figure 9*). Unlike many other models of hepatic steatosis, there were no significant alterations in lipogenic gene expression either in the fed or fasted state with *Mboat7* knockdown (*Figure 9A*). Unexpectedly, the expression of carnitine palmitoyl transferase 1 (Cpt1) was significantly elevated with *Mboat7* knockdown (*Figure 9A*), but this would not be expected to promote the accumulation of neutral lipids. The expression of enzymes involved in cholesterol biosynthesis (*Hmgcr* and *Hmcgs1*) were modestly reduced in *Mboat7* ASO-treated mice, but only in the fasted state (*Figure 9A*). In addition, expression of the cholesterol efflux regulator *Abca1* was elevated in *Mboat7* ASO-treated mice, but only in the fasted state. Altogether, these minor differences in hepatic gene expression are unlikely to be driving the lipid accumulation in Mboat7 ASO treated mice. Next, we evaluated whether *Mboat7* may influence the export of neutral lipids via packaging on nascent very low density lipoproteins (VLDL). However, *Mboat7* ASO-treated mice did not have significant differences in VLDL-TG secretion during a detergent block (*Figure 9B*). Furthermore, steady state plasma levels of triglycerides and total cholesterol were not significantly altered in *Mboat7* ASO-treated mice other than a minor reduction in cholesterol content in both low density lipoproteins (LDL) and high density lipoproteins (HDL) only in chow-fed cohorts (*Figure 2—figure supplement 5*). Another common cause of fatty liver is increased delivery of adipose-derived fatty acids to the liver, as is commonly seen with prolonged fasting and certain types of lipodystrophies. However, *Mboat7* knockdown did not alter basal or catecholamine-stimulated adipocyte lipolysis in vivo (*Figure 9C*), ruling out a role for altered adipose lipolysis as a contributing factor. Finally, we examined the possibility that MBOAT7 may be a determinate of metabolism locally at the surface of cytosolic lipid droplets to regulate hepatic steatosis, given that one recent study reported that MBOAT7 can localize to cytosolic lipid droplets

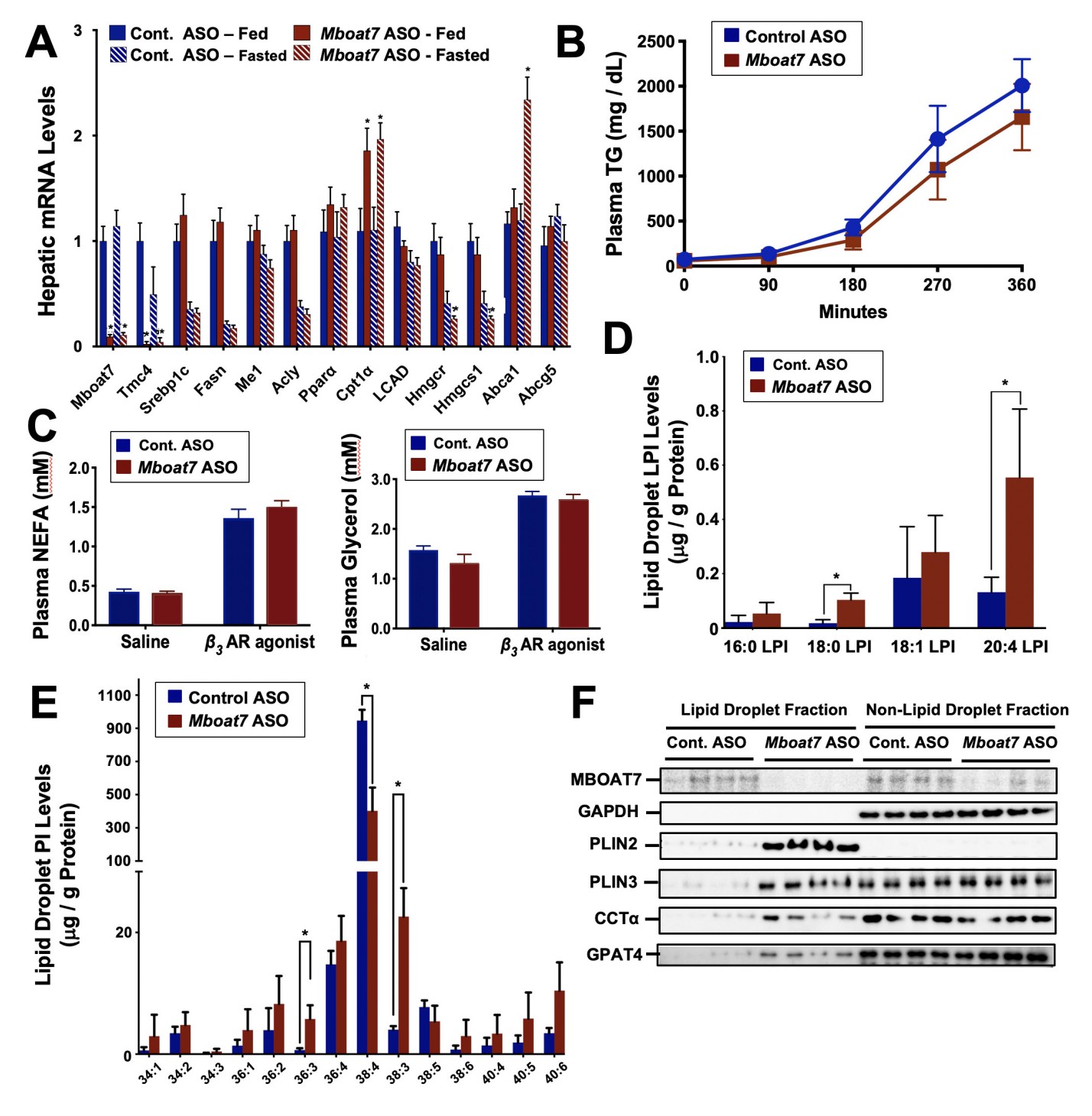

**Figure 9.** Mboat7 knockdown promotes hepatic steatosis by altering the lipidome and proteome of the cytosolic lipid droplets. Male C57BL/6 mice were fed either chow or a HFD for 12–20 weeks along with receiving a control non-targeting ASO or an ASO targeting the knockdown of *Mboat7*. (A) Mice were necropsied either in the fed or fasted (12 hr fast) state, and the hepatic expression of genes involved in lipogenesis, fatty acid oxidation, or sterol sensing analyzed using qPCR. (B) Plasma triglyceride (TG) secretion rate in mice administered Triton WR-1339 (tyloxapol). (C) Fed mice were treated with either saline vehicle or the β3-adrenergic receptor agonist CL316243 (0.1 mg/g body weight) to stimulate adipocyte lipolysis, and plasma glycerol and NEFA were measured after 15 min. Panels D–F), ASO-treated mice were fasted for 4 hr (from 9:00 a.m. to 1:00 p.m.) prior to necropsy, and hepatic lipid droplets were isolated using sucrose gradient fractionation to analyze both the lipidome and proteome of cytosolic lipid droplets. Lipid droplet fractions were extracted and analyzed by LC-MS/MS to quantify either lysophosphatidylinositol (D) or phosphatidylinositol (E) species on isolated lipid droplets. (F) Lipid droplet fractions were analyzed using Western blotting for MBOAT7, glycerol 3-phosphate dehydrogenase (as a non-

*Figure 9 continued on next page*

Figure 9 continued

lipid droplet fraction control), perilipin 2 (PLIN2), perilipin 3 (PLIN3), CTP:phosphocholine cytidylyltransferase α (CCTα), or glycerol-3-phosphate 4 (GPAT4); four individual mice per group are shown. Data are presented as mean ± S.E.M, (n = 4–6 per group; *p≤0.05, **p≤0.01, ***p≤0.001, ****p≤0.0001; Two-way ANOVA with Tukey's *post-hoc* test).
DOI: https://doi.org/10.7554/eLife.49882.026

(*Mancina et al., 2016*). We confirmed that MBOAT7 can indeed be found in lipid droplets isolated by sucrose gradient fractionation (*Figure 9F*), and plays a regulatory role in both the lipidome and proteome of cytosolic lipid droplets. Cytosolic lipid droplets isolated from *Mboat7* ASO-treated mice showed marked accumulation of 18:0 LPI and 20:4 LPI, but not 16:0 LPI or 18:1 LPI (*Figure 9D*). This is in stark contrast to what is seen in the whole liver, where *Mboat7* knockdown instead promotes accumulation of 16:0 LPI and 18:1 LPI (*Figure 6D*). Also, lipid droplets isolated from *Mboat7* ASO-treated mice have a reduced level of 38:4 PI but a reciprocal increase in 36:3 and 38:3 PI (*Figure 9E*), which is quite different when to compared to effects in whole liver (*Figure 6E*). In addition to alterations in the lipid droplet lipidome, *Mboat7* knockdown is associated with accumulation of several proteins involved in lipid synthesis and storage on isolated lipid droplets (*Figure 9F*). Both perilipin 2 and 3 (PLIN2 and PLIN3) were much more abundant on lipid droplets isolated from *Mboat7* ASO-treated mice, as were the critical lipid synthetic enzymes CTP:phosphocholine cytidylyltransferase α (CCTα) and glycerol-3-phosphate 4 (GPAT4). Given that recent reports have shown that lipid droplet targeting of CCTα and GPAT4 are critical regulators of the overall size and triglyceride storage capacity of lipid droplets (*Guo et al., 2008*; *Krahmer et al., 2011*; *Wilfling et al., 2013*), these proteomic alterations at the lipid droplet surface may to contribute to the hepatic steatosis seen in *Mboat7* ASO-treated mice. In order to understand cell autonomous effects of MBOAT7 lipid metabolism we generated MBOAT7-deficient cells via CRISPR-Cas9-mediated genome editing. Huh7 lacking MBOAT7 have increased lipid droplets upon fatty acid loading (*Figure 10A,B*). This lipid accumulation was due in part to increases in de novo lipogenesis rates (*Figure 10C*), and reciprocal decreases in fatty acid oxidation rates (*Figure 10D*). However, MBOAT-deficient Huh7 cells did not have altered turnover of stored triacylglycerol (*Figure 10E*) or cholesterol ester (*Figure 10F*).

## Discussion

Given that several recent studies have found a strong link between the common rs641738 variant allele and liver disease progression, there is considerable interest in identifying the causative gene within this locus. Although the rs641738 polymorphisms maps to the first exon of the poorly annotated gene *TMC4*, this study is the first to demonstrate that genetic deletion of *Tmc4* does not result in hepatic steatosis. Instead, selective loss of function of the neighboring gene *Mboat7* is sufficient to sensitize mice to high fat diet-driven liver disease progression. The major findings of the current study include the following: (1) Genetic deletion of *Tmc4* does not alter hepatic lipid storage, (2) Hepatic expression of *MBOAT7* is reduced in obese humans and rodents, independent of rs641738 status, (3) *Mboat7* expression in mouse liver and adipose tissue is negatively correlated with obesity and insulin sensitivity across the strains represented in the Hybrid Mouse Diversity Panel, (4) *Mboat7* knockdown promotes hepatic steatosis, hepatocyte death, inflammation, and early gene expression profiles consistent with fibrosis, but this only happens when mice are challenged with a high fat diet, (5) *Mboat7* loss of function promotes striking hyperinsulinemia and hepatic insulin resistance, (6) *Mboat7* knockdown results in tissue-specific reorganization of its substrates (LPI) and product (PI) lipids, (7) LPI lipids can rapidly induce hepatic inflammatory and fibrotic gene expression programs in an *Mboat7*-dependent manner in mice, and (8) the hepatic steatosis seen with *Mboat7* knockdown is not related to differences in lipogenic gene expression, VLDL secretion, or tissue lipolysis, but instead may be related to alterations in hepatic lipid droplet accumulation of lipogenic enzymes (CCTα and GPAT4) that allow for expansion of large lipid droplets (*Guo et al., 2008*; *Krahmer et al., 2011*; *Wilfling et al., 2013*), and increased rates of de novo lipogenesis and reduced fatty acid oxidation. Collectively, these data support a role for *Mboat7*-driven acylation of LPI lipids as a key protective mechanism against obesity-linked NAFLD progression.

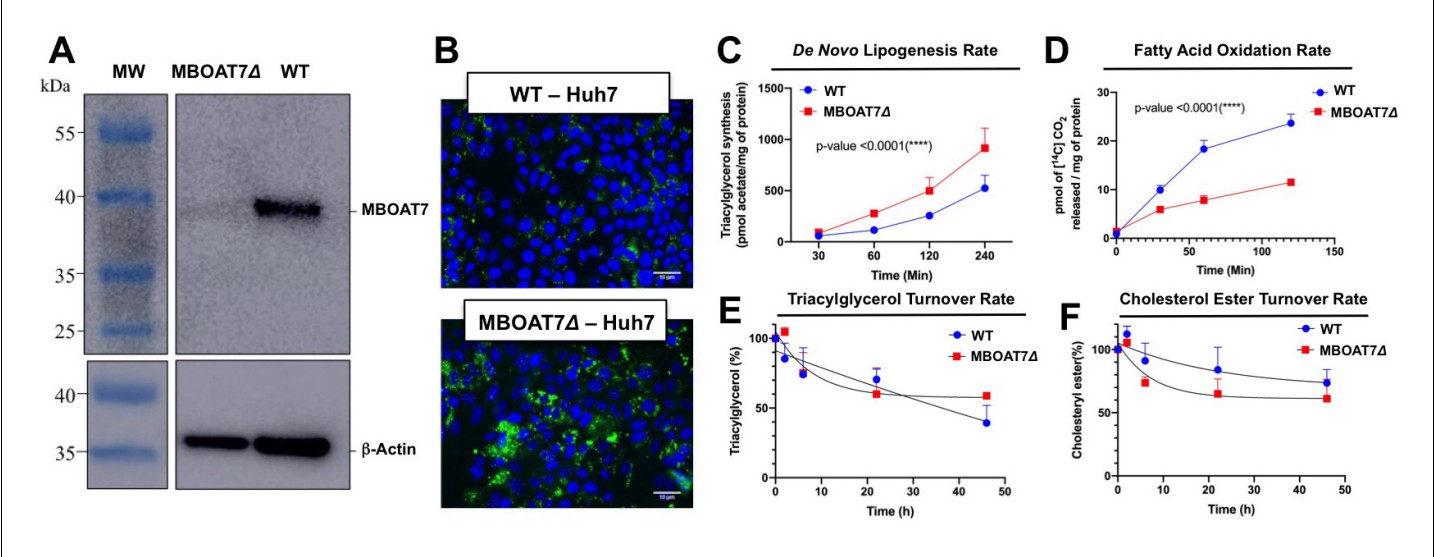

**Figure 10.** Genetic deletion of MBOAT7 in Huh7 cells results in cell autonomous alterations in lipogenesis and fatty acid oxidation. CRISPR-Cas9 gene editing was used to delete the *MBOAT7* gene from Huh7 hepatoma cells: (**A**), Immunoblot confirmation of genetic deletion of *MBOAT7* (*MBOAT7Δ*): Microsome were isolated from WT and *MBOAT7Δ*cells, and MBOAT7 protein levels were detected via Western blotting. (**B**) Fluorescent microscopy image of control and *MBOAT7Δ*-Huh7 cells were supplemented with 400 μM oleic acid for 24 hr. Following 24 hr of lipid loading, cells were washed with PBS and stained with Bodipy (493/503). Scale bar = 10 μm. (**C**) Effect of *MBOAT7Δ* on de novo lipogenesis: Control and *MBOAT7Δ*-Huh7 cells were incubated [$^{14}$C]acetate in the presence of broad lipase inhibitors. Cells were harvested at various time point, lipids extracted and the incorporation of [$^{14}$C]acetate into [$^{14}$C]triacylglycerol. (**D**) Effect of *MBOAT7Δ* on complete fatty acid oxidation: The graph shows the amount of $^{14}$C-palmitic acid oxidized to $^{14}$C-CO (*Mancina et al., 2016*). (**E–F**), Effect of *MBOAT7Δ* on triacylglycerol (**E**) and cholesterol ester (**F**) hydrolysis: Control and *MBOAT7Δ*-Huh7 cells were incubated with 100 μM oleate complexed to bovine serum albumin + 1 μCi[$^{3}$H]-cholesterol + 0.5 μCi-[$^{14}$C]-oleate for 24 hr to label cellular cholesteryl ester and triacylglycerol pools to steady state. The supplemental fatty acids were withdrawn, and the cells were incubated with 6 μM triacsin C to inhibit triacylglycerol synthesis and re-esterificaiton. Cells were harvested, lipids were extracted and separated by thin layer chromatography to determine the turnover of cholesteryl esters and triacylglycerols. Data shown are mean ± S.E.M. and are representative of two separate experiments; ***p<0.0002, ****p<0.0001; two-way ANOVA with Bonferroni's multiple comparisons test.

DOI: https://doi.org/10.7554/eLife.49882.027

The MBOAT family of enzymes are critical players in determining the composition of fatty acids in cellular membranes, and are emerging as key players in cardiometabolic disease (*Rong et al., 2013*; *Harayama et al., 2014*). It is important to note that several other lysophospholipid acyltransferase enzymes have been linked to inflammatory diseases (*Rong et al., 2013*; *Harayama et al., 2014*). In particular, lysophosphatidylcholine acyltransferases (LPCAT1, LPCAT2, and LPCAT3) are known to regulate tissue inflammation and endoplasmic reticulum stress by altering arachidonic acid availability as well as determining membrane phosphatidylcholine saturation (*Rong et al., 2013*; *Harayama et al., 2014*). In fact, the lysophospholipid acyltransferase family sits at a critical signaling nexus, given that they can play key roles in the generation of arachidonic acid-derived lipid mediators as well as regulating the levels of lysophospholipid signaling lipids (*Shindou and Shimizu, 2009*; *Shindou et al., 2009*). Given the fact that MBOAT7 preferentially generates PUFA-enriched PIs it also has the unique potential to also impact phosphorylated PI species (PtdInsP, PtdInsP2, and PtdInsP3, etc.) (*Anderson et al., 2013*). In agreement with our findings with PI lipids (**Fil**) in *Mboat7* ASO-treated mice, global genetic deletion of *Mboat7* selectively lowers the 38:4 molecular species of PIPs, while increasing more saturated species (*Anderson et al., 2013*). Therefore, we cannot rule out that alterations in PIP-dependent signal transduction may also play a role in MBOAT7 loss of function-driven liver disease progression.

Unexpectedly, *Mboat7* knockdown did not significantly alter hepatic lipid storage or inflammation in chow-fed mice. Instead there is a clear unmasking of *Mboat7*-dependent fatty liver phenotypes only when mice are fed with a high fat diet. This could potentially indicate that unknown dietary factors play a regulatory role and limit hepatic LPIAT activity via unknown mechanisms (*Figure 6A*). Our data also support the idea that when mice, rats, and humans become obese *MBOAT7* expression is

reduced to varying degrees (*Figure 1A–C*), indicating regulation either at the transcriptional of post-transcriptional level. Future work should focus on understanding how dietary factors and adiposity itself impacts *MBOAT7* expression and activity, and under what dietary and BMI conditions are *MBOAT7* polymorphisms predicted to be most deleterious. In humans, the rs641738 variant has been clearly linked to end stage fibrotic liver disease (*Buch et al., 2015*; *Mancina et al., 2016*; *Luukkonen et al., 2016*; *International Liver Disease Genetics Consortium et al., 2016*). However, an important limitation of our studies is that high fat diet feeding in mice is not sufficient to drive bridging fibrosis. In future studies, it will be important to examine how *Mboat7* loss of function impacts the development of fibrosis in appropriate fibrosis-prone animal models, and whether *Mboat7* expression in hepatic stellate cells plays a regulatory role in the progression from NASH to cirrhosis. From this study it is clear that MBOAT7 can diversify the inositol-containing phospholipids and the associated proteome on cytosolic lipid droplets, and this could in part explain the large lipid droplets that accumulate in *Mboat7* ASO-treated mice. It is interesting to note that the well-known PNPLA3 variant associated with fatty liver disease (I148M) accumulates on lipid droplets, and similarly reorganizes the lipidome and proteome to promote liver disease progression (*BasuRay et al., 2019*; *Wang et al., 2019*; *Mitsche et al., 2018*; *BasuRay et al., 2017*). Our findings here with MBOAT7-driven restructuring of the lipid droplet surface, and those recently published with PNPLA3 (*BasuRay et al., 2019*), suggest that alterations in lipid modifying enzyme access to the surface of cytosolic lipid droplets may be a common mechanism by which human fatty liver develops. Furthermore, we show that genetic deletion of *MBOAT7* in Huh7 hepatoma cells can promote cell autonomous increases in cytosolic lipid droplets, and this enhanced lipid storage is associated with augmented lipogenesis rates and reduced fatty acid oxidation rates. Our studies provide new clues into new therapeutic leads for advanced liver diseases.

Another particularly striking finding from the current study is that *Mboat7* knockdown promotes severe hyperinsulinemia (*Figure 5B*). Based on our results examining both insulin and C-peptide release during a glucose tolerance test (*Figure 5B,C*), it is reasonable to assume that *Mboat7* knockdown is enhancing glucose-stimulated insulin secretion (GSIS) in pancreatic β cells. However, it is important to note that *Mboat7* ASO treatment did not alter *Mboat7* expression (data not shown) or LPI/PI lipids in the pancreas (*Figure 6—figure supplement 4*). Therefore, it is tempting to speculate that ASO-mediated knockdown of *Mboat7* facilitates the production of an endocrine signaling lipid (possibly LPI) that may impact β cell GSIS. Interestingly, there are reports demonstrating that LPI lipids can stimulate β cell GSIS (*Metz, 1986*; *Metz, 1988*). It is interesting to note that expression of *Mboat7* in white adipose tissue is modestly negatively correlated with HOMA-IR in the hybrid mouse diversity panel (*Figure 1F*), yet ASO-mediated knockdown of *Mboat7* did not alter insulin signaling in adipose tissue (*Figure 5F*). These data suggest that additional studies are needed to clarify a potential role for Mboat7 in adipose tissue insulin sensitivity. Collectively, our data suggest that under conditions where MBOAT7 activity is suppressed (obesity or with the rs641738 variant) inefficient acylation of LPI substrate allows for these lipids to accumulate and initiate autocrine, paracrine, and potentially endocrine signaling that impact the progression of NAFLD and insulin resistance. Therefore, further investigation into the receptor system(s) that sense LPI lipids could hold therapeutic promise in liver disease and other associated metabolic diseases. Collectively, these studies identify MBOAT7-driven acylation of LPI lipids as an important modulator of both liver disease progression and associated type 2 diabetes.

## Materials and methods

**Key resources table**

| Reagent type (species) or resource | Designation | Source or Reference | Identifiers | Additional Information |
|---|---|---|---|---|
| Genetic reagent (*M. musculus*) | C57BL/6NJ - C57BL/6NJ-*Tmc4*$^{em1(IMPC)J}$/Mmjax | Jackson Laboratory | Stock #: 46062-JAX; RRID: MGI:5882504; RRID: IMSR_JAX:032275 | |

*Continued on next page*

*Continued*

| Reagent type (species) or resource | Designation | Source or Reference | Identifiers | Additional Information |
|---|---|---|---|---|
| Sequence-based reagent | Mboat7 | Sigma | | F: CATGCGGTACTGGAACATGA<br>R: CCAGTAGGCGCTCAGCAG |
| Sequence-based reagent | Cyclophilin A | Sigma | | F: CCTGCTCTCCTCTCACCTCCT<br>R: TAGAAGCGCATGCGGAAGG |
| Sequence-based reagent | Mboat7 | Sigma | | F: atggcatgcgatactggaac<br>R: agcatggtccaggcactc |
| Sequence-based reagent | Tmc4 | Sigma | | F: CGGTACATCCACAAACATGG<br>R: GAGATCGTTCAGCACAGACG |
| Sequence-based reagent | Tnfα | Sigma | | F: CCACCACGCTCTTCTGTCTAC<br>R: AGGGTCTGGGCCATAGAACT |
| Sequence-based reagent | IL-1β | Sigma | | F: AGTTGACGGACCCCAAAAG<br>R: AGCTGGATGCTCTCATCAGG |
| Sequence-based reagent | Col1a2 | Sigma | | F: GCAGGTTCACCTACTCTGTCCT<br>R: CTTGCCCCATTCATTTGTCT |
| Sequence-based reagent | α-SMA | Sigma | | F: GTCCCAGACATCAGGGAGTAA<br>R: TCGGATACTTCAGCGTCAGGA |
| Sequence-based reagent | Cd11c | Sigma | | F: gagccagaacttcccaactg<br>R: tcaggaacacgatgtcttgg |
| Sequence-based reagent | Srebp1c | Sigma | | F: tctcactccctctgatgctac<br>R: gcaaccactgggtccaatta |
| Sequence-based reagent | Col1a1 | Sigma | | F: ATGTTCAGCTTTGTGGACCTC<br>R: CAGAAAGCACAGCACTCGC |
| Sequence-based reagent | Fas | Sigma | | F: GCTGCGGAAACTTCAGGAAAT<br>R: AGAGACGTGTCACTCCTGGACTT |
| Sequence-based reagent | Acc1 | Sigma | | F: CCTGAGGAACAGCATCTCTAAC<br>R: GCCGAGTCACCTTAAGTACATATT |
| Sequence-based reagent | Scd1 | Sigma | | F: TTCCCTCCTGCAAGCTCTAC<br>R: CAGAGCGCTGGTCATGTAGT |
| Sequence-based reagent | Cpt1a | Sigma | | F: TCCATGCATACCAAAGTGGA<br>R: TGGTAGGAGAGAGCAGCACCTT |
| Sequence-based reagent | Cgi58 | Sigma | | F: gcggtgatgaaagcgatg<br>R: caccctgtcagccatcctg |
| Sequence-based reagent | Atgl | Sigma | | F: CTTCCTCGGGGTCTACCACA<br>R: GCCTCCTTGGACACCTCAATAA |
| Sequence-based reagent | Cd68 | Sigma | | F: ATCCCCACCTGTCTCTCTCA<br>R: ACCGCCATGTACTCCAGGTA |
| Sequence-based reagent | F4/80 | Sigma | | F: GGATGTACAGATGGGGGATG<br>R: CATAAGCTGGGCAAGTGGTA |
| Sequence-based reagent | Desmin | Sigma | | F: GTGGATGCAGCCACTCTAGC<br>R: TTGCCGCGATGGTCTCATAC |
| Sequence-based reagent | Rgs16 | Sigma | | F: AGGGCTCACCACATCTT<br>R: AGGTTTGTCTTGGTCAGTTC |
| Sequence-based reagent | Lcn2 | Sigma | | F: ccatctatgagctacaagagaacaat<br>R: tctgatccagtagcgacagc |
| Sequence-based reagent | Saa1 | Sigma | | F: CCAGGATGAAGCTACTCACCA<br>R: TAGGCTCGCCACATGTCC |
| Sequence-based reagent | Saa2 | Sigma | | F: ACTATGATGCTGCCCAAAGG<br>R: CTCTGCCGAAGAATTCCTGAAA |
| Sequence-based reagent | Ly6d | Sigma | | F: CAACTGTAAGAACCCTCAGGTC<br>R: CACTCTTTCCTCACCAGGTTC |
| Sequence-based reagent | Onecut1 | Sigma | | F: atcctcatgcccacctga<br>R: cctgaattacttccattgctga |

*Continued on next page*

*Continued*

| Reagent type (species) or resource | Designation | Source or Reference | Identifiers | Additional Information |
|---|---|---|---|---|
| Sequence-based reagent | Fasn | Sigma | | F: GTCACCACAGCCTGGACCGC<br>R: CTCGCCATAGGTGCCGCCTG |
| Sequence-based reagent | Me1 | Sigma | | F: ggagctccaggtccttagaata<br>R: tctgtcttgcaggtccattaac |
| Sequence-based reagent | Acly | Sigma | | F: CTCACACGGAAGCTCATCAA<br>R: TCCAGCATTCCACCAGTATTC |
| Sequence-based reagent | Pparα | Sigma | | F: GGAGGCGTTTCCTGAGACC<br>R: CAGCCACAAACGTCAGTTCAC |
| Sequence-based reagent | LCAD | Sigma | | F: ccggttctttgaggaagtgaa<br>R: agtgtcgtcctccaccttctc |
| Sequence-based reagent | Hmgcr | Sigma | | F: CTTGTGGAATGCCTTGTGATTG<br>R: AGCCGAAGCAGCACATGAT |
| Sequence-based reagent | Hmgcs1 | Sigma | | F: GCCGTGAACTGGGTCGAA<br>R: GCATATATAGCAATGTCTCCTGCAA |
| Sequence-based reagent | Abca1 | Sigma | | F: GGGCTGCCACCTCCTCAGAGAAA<br>R: CACATCCTCATCCTCGTCATTC |
| Sequence-based reagent | Abcg5 | Sigma | | F: TCCTGCATGTGTCCTACAGC<br>R: ATTTGCCTGTCCCACTTCTG |
| Sequence-based reagent | CycloA | Sigma | | F: gcggcaggtccatctacg<br>R: gccatccagccattcagtc |
| Sequence-based reagent | MBOAT7 | Sigma | *MBOAT7-E5-Nick-5F* | CACCGTCCATCAGGGAGGGCACGTC |
| Sequence-based reagent | MBOAT7 | Sigma | *MBOAT7-E5-Nick-5R* | AAACGACGTGCCCTCCCTGATGGAC |
| Sequence-based reagent | MBOAT7 | Sigma | *MBOAT7-E5-Nick-3F* | CACCGCAGCTACAGCTACTGCTACG |
| Sequence-based reagent | MBOAT7 | Sigma | *MBOAT7-E5-Nick-3R* | AAACGTAGCAGTAGCTGTAGCTGC |
| Antibody | Anti-MBOAT7 (Rat monoclonal) | PMID: 23097495 | RRID:AB_2813851 | 1:1000 |
| Antibody | Anti-pIRβ$^{Y1150/1151}$(Rabbit monoclonal) | Cell Signaling | RRID: AB_331253 | 1:1000 |
| Antibody | Anti-TMC4 (Rabbit polyclonal) | Thermo Fisher | Cat#: OSR00225W; RRID: AB_2204190 | 1:500 |
| Antibody | Anti-IRβ (Rabbit monoclonal) | Cell Signaling | Cat#: 3025S; RRID: AB_2280448 | 1:1000 |
| Antibody | Anti-pAkt$^{S473}$(rabbit polyclonal) | Cell Signaling | Cat#: 9271S; RRID: AB_329825 | 1:1000 |
| Antibody | Anti-pAkt$^{T308}$ (Rabbit monoclonal) | Cell Signaling | Cat#: 13038S; RRID: AB_2629447 | 1:1000 |
| Antibody | Anti-Akt (Rabbit polyclonal) | Cell Signaling | Cat#: 9272S; RRID: AB_329827 | 1:1000 |
| Antibody | Anti-PLIN2 (Rabbit polyclonal) | Novus Biologicals | Cat#: NB110-40877; RRID: AB_787904 | 1:1000 |
| Antibody | Anti-PLIN3 (Rabbit polyclonal) | Novus Biologicals | Cat#: NB110-40764; RRID: AB_715116 | 1:1000 |
| Antibody | Anti-CCTα (Rabbit monoclonal) | Cell Signaling | Cat#: 6931; RRID: AB_10830058 | 1:1000 |
| Antibody | Anti-GPAT4 (Rabbit Polyclonal) | Novus Biologicals | Cat#: NB100-2390; RRID: AB_2273811 | 1:1000 |
| Antibody | Anti-GAPDH-HRP (Rabbit monoclonal) | Cell Signaling | Cat#: 8884; RRID: AB_11129865 | 1:5000 |
| Antibody | Anti-β-Actin-HRP (Mouse monoclonal) | Proteintech | Cat#: HRP-60008; RRID: AB_2289225 | 1:5000 |

*Continued on next page*

*Continued*

| Reagent type (species) or resource | Designation | Source or Reference | Identifiers | Additional Information |
|---|---|---|---|---|
| Antibody | Anti-CD3e-APC-Cy7 (Hamster monoclonal) | BD Pharm; clone: 145–2 C11 | Cat#: 557596; RRID: AB_396759 | 1:100 |
| Antibody | Anti-CD4-Alexa Fluor 700 (Rat monoclonal) | BD Pharm; clone: RM4-5 | Cat#: 557308; RRID: AB_396634 | 1:100 |
| Antibody | Anti-CD8a-PE-Cy7 (Rat monoclonal) | eBiosciences; clone: 53–6.7 | Cat#: 25-0081-81; RRID: AB_469583 | 1:100 |
| Antibody | Anti-Cd11b-PE-Cy7 (Rat monoclonal) | BD Pharm | Cat#: 552850; RRID: AB_394491 | 1:100 |
| Antibody | Anti-Cd11c-PE-CF 594 (Hamster monoclonal) | BD Pharm | Cat#: 562454; RRID: AB_2737617 | 1:100 |
| Antibody | Anti-CD206 Alexa Fluor 647 (Rat monoclonal) | Biolegend | Cat#: 141712; RRID: AB_10900420 | 1:100 |
| Antibody | Anti-rabbit IgG HRP (Donkey) | GE-Healthcare | Cat#: NA934-100UL; RRID: AB_772206 | 1:5000 |
| Antibody | Anti-rat IgG HRP (Goat) | Santa Cruz | Cat#: SC-2006; RRID: AB_1125219 | 1:5000 |
| Antibody | IRDye 800 anti-Rabbit IgG (Goat) | LiCor | Cat#: 926–32211; RRID: AB_621843 | 1:10,000 |
| Sequence-based reagent | Mboat7 | Ionis | | |
| Commercial assay or kit | Supersignal West Pico Plus substrate | Thermo Fisher | 34577 | |
| Commercial assay or kit | AST kit | Sekisui Diagnostics | 319–30 | |
| Commercial assay or kit | ALT kit | Sekisui Diagnostics | 318–30 | |
| Commercial assay or kit | Liver TG | Wako | 994–02891 | |
| Commercial assay or kit | Free Cholesterol | Wako | 993–02501 | |
| Commercial assay or kit | Phospholipid C | Wako | 433–36201 | |
| Commercial assay or kit | Total Cholesterol | Fisher Sci | TR134321 | |
| Commercial assay or kit | Insulin ELISA | Millipore | EZRMI-13K | |
| Commercial assay or kit | C-Peptide ELISA | Crystal Chem | 90050 | |
| Chemical compound, drug | 16:0 LPI | Avanti Polar Lipid, Inc | 850102P | 50 µM |
| Chemical compound, drug | 18:0 LPI | Avanti Polar Lipid, Inc | 850104P | 50 µM |
| Chemical compound, drug | 18:1 LPI | Avanti Polar Lipid, Inc | 850100P | 50 µM |
| Chemical compound, drug | Arachidonyl Coenzyme A [arachidonyl-1–14C]; 50–60 mCi/mmol 1.85–2.22 GBq/mmol | American Radiolabeled Chemicals | ARC 0519 | 0.025 µCi |
| Other | TLC silica gel Plate 60 F254 | Millipore Triton WR-1339 | 1055540001 | reagent |

*Continued on next page*

*Continued*

| Reagent type (species) or resource | Designation | Source or Reference | Identifiers | Additional Information |
|---|---|---|---|---|
| Chemical compound, drug | Beta-3 AR Agonist | Sigma | C5976-5mg | |
| Chemical compound, drug | Tyloxapol | Sigma | T0307-10G | 500 mg/kg |
| Peptide, recombinant protein | Insulin | Sigma | I2643 | 0.35 U/kg |

## Human *MBOAT7* expression levels in lean and obese subjects

The majority of subjects recruited to examine MBOAT7 expression levels were morbidly obese bariatric surgery patients, but we were able to obtain liver biopsies from 10 subjects with a BMI under 30 as normal weight controls. For recruitment, adult patients undergoing gastric bypass surgery at Wake Forest School of Medicine were consented via written consent and enrolled by a member of the study staff following institutionally approved IRB protocols as previously described (*Shores et al., 2011*). Exclusion criteria included: positive hepatitis C antibody, positive hepatitis B surface antigen, history of liver disease other than NAFLD, Childs A, B, or C cirrhosis, past or present diagnosis/treatment of malignancy other than non-melanocytic skin cancer, INR greater than 1.8 at baseline or need for chronic anticoagulation with warfarin or heparin products, use of immunomodulation for or history of inflammatory diseases including but not limited to malignancy, rheumatoid arthritis, psoriasis, lupus, sarcoidosis and inflammatory bowel disease, and greater or equal to seven alcohol drinks per week or three alcoholic drinks in a given day each week. In addition to bariatric surgery patients, a small number of non-obese subjects (body mass index <30.0) consented to liver biopsy during elective gall bladder removal surgery (n = 10). Each subject was assigned a unique identifier which was used throughout the study and did not include any identifiable information about the patient such as name, address, telephone number, social security number, medical record number or any of the identifiers outlined in the HIPAA Privacy Rule regulations. Only the principal investigator had access to the code linking the unique identifier to the study subject. Basic clinical information was obtained via self-reporting and a 15 ml baseline blood sample was obtained at the time of enrollment. A subset of this cohort has been previously described (*Schugar et al., 2017*; *Shores et al., 2011*). At the time of surgery, the surgeon collected a roughly 1-gram sample from the lateral left lobe. Wedge biopsies were rinsed with saline and immediately snap frozen in liquid nitrogen in the operating room before subsequent storage at −80℃. For data shown in *Figure 7* showing levels of MBOAT7 substrate and product lipids, de-identified patient samples from the Cleveland Clinic hepatology clinic (IRB # 10–947) were analyzed. These patients had biopsy proven Ishak fibrosis scores (*Ishak et al., 1995*) of 0 (normal) or 4 (advanced fibrosis). For analysis of hepatic *MBOAT7* expression, RNA isolated from liver biopsies were used for quantitative real time PCR (qPCR) as described below.

## Rat studies of diet-induced obesity

Sprague Dawley Rats were received at 12 weeks of age and were housed in individual cages, kept at a constant temperature and ambient humidity in a 12-h light/dark cycle. Animals were then randomly assigned to either a standard chow diet or a high-fat diet (D12492, 60% fat, Research Diets, New Brunswick, NJ, USA) ad libitum to establish diet-induced obesity as previously described (*Schugar et al., 2017*). After 6 months of HFD-feeding, livers were excised for standard qPCR analysis of *Mboat7* expression.

## Hybrid Mouse Diversity Panel

92 inbred strains of 8-week-old male mice (180 individual mice) were fed a high fat, high sucrose diet (D12266B, Research Diets, New Brunswick, NJ) for 8 weeks before tissue collection (*Parks et al., 2013*). Gene expression of *Mboat7* in white adipose tissue and liver were measured and correlated with obesity related traits using biweight midcorrelation analysis as previously described (*International Liver Disease Genetics Consortium et al., 2016*).

## Mouse studies of Mboat7 loss of function

To explore the role of *Mboat7* in diet-induced obesity, NAFLD progression, and insulin resistance, we utilized an in vivo knockdown approach in 8 week old adult mice. Selective knockdown of *Mboat7* was accomplished using 2'-O-ethyl (cET) modified antisense oligonucleotides (ASO). All ASOs used in this work were synthesized, screened, and purified as described previously (*Crooke et al., 2005*) by Ionis Pharmaceuticals, Inc (Carlsbad, CA). For *Mboat7* knockdown studies, adult (8 week old) male C57BL/6 mice were purchased from Jackson Labs (Bar Harbor, ME USA), and maintained on either a standard rodent chow diet or a high fat diet (HFD, D12492 from Research Diets Inc) and injected intraperitoneally biweekly with 12.5 mg/kg of either non-targeting control ASO or one of two independent ASOs directed against murine *Mboat7* for a period of 20 weeks. Similar results were seen with two independent ASOs targeting different regions of the *Mboat7* mRNA, hence key data using one *Mboat7* ASO are shown. All mice were maintained in an Association for the Assessment and Accreditation of Laboratory Animal Care, International-approved animal facility, and all experimental protocols were approved by the Institutional Animal Care and use Committee of the Cleveland Clinic (IACUC protocols # 2015–1519 and # 2018–2053).

## Studies in global *Tmc4* knockout mice

Global *Tmc4* knockout mice (C57BL/6NJ-*Tmc4*[em1(IMPC)J]/Mmjax) were provided by the Knockout Mouse Phenotyping Program (KOMP) at The Jackson Laboratory using CRISPR technology. Briefly, guide RNAs (GGAACCAGACCTTTTCCCAA and GAGTCAGCGTCAGAAAATGA) were designed to insert create a 277 bp deletion in exon 3 of the transmembrane channel-like gene family 4 (*Tmc4*) gene beginning at Chromosome 7 position 3,675,326 bp and ending after 3,675,602 bp (GRCm38/mm10). The mutation is predicted to delete ENSMUSE00001301550 (exon 3) and 131 bp of flanking intronic sequence, including the splice acceptor and donor, and is predicted to cause a change of amino acid sequence after residue 87 and early truncation six amino acids later. Guide RNAs and Cas9 nuclease were introduced into C57BL/6NJ-derived fertilized eggs with well recognized pronuclei. Embryos were transferred to pseudopregnant females. Correctly targeted pups were identified by sequencing and further bred to C57BL/6NJ (Stock No. 005304) to develop the colony. Once the stock mice arrived at the Cleveland Clinic heterozygous mice were intercrossed to generate additional heterozygous (*Tmc4*[+/-]) and homozygous wild type (*Tmc4*[+/+]) and knockout (*Tmc4*[-/-]) progeny. In addition to wild type mice on the mixed C57BL/6NJ background, we also studied parallel wild type mice on a pure C57BL/6J strain to increase sample size of experimental controls. Experimental mice were fed a high fat diet for 2 weeks prior to necropsy for analysis of hepatic steatosis.

## Standardized necropsy conditions

To keep results consistent, the vast majority of experimental mice were fasted for 4 hr (from 9:00 a.m. to 1:00 p.m.) prior to necropsy. For the fasting versus fed experiments, fed mouse tissue were collected at the beginning of the light cycle (7:00 a.m.) in ad libitum fed mice, whereas the fasted group had food removed at 7:00 p.m. and were necropsy after a 12 hr fast (7:00 a.m.). At necropsy, all mice were terminally anesthetized with ketamine/xylazine (100–160 mg/kg ketamine-20–32 mg/kg xylazine), and a midline laparotomy was performed. Blood was collected by heart puncture. Following blood collection, a whole body perfusion was conducted by puncturing the inferior vena cava and slowly delivering 10 ml of saline into the heart to remove blood from tissues. Tissues were collected and immediately snap frozen in liquid nitrogen for subsequent biochemical analysis or fixed for morphological analysis.

## Plasma ALT and AST analysis

To determine the level of hepatic injury in mice fed chow and HFD with ASO treatment for 20 weeks, plasma was used to analyze aminotransferase (AST) and alanine aminotransferase (ALT) levels using enzymatic assays (Sekisui Diagnostics, Lexington, MA, USA).

## In vivo measurement of adiposity and lean mass

Quantitation of lean and fat mass were done using an EchoMRITM-130 Body Composition Analyzer (EchoMRI International).

## Isolation and characterization of lipid droplets from mouse liver

Hepatic LDs were isolated by sucrose gradient centrifugation as we have previously described (*Ferguson et al., 2017*). Approximately 100 mg of tissue was minced with a razor blade on a cold surface. Minced tissue was transferred to a Potter-Elvehjem homogenizer, and then 200 µl of 60% sucrose was added to the tissue sample and incubated on ice for 10 min. Next, 800 µl of lysis buffer was added and mixed, and then incubated on ice for 10 min. Samples were homogenized with five strokes of a Teflon pestle and transferred to a 2 ml centrifuge tube. Lysis buffer (600 µl) was carefully layered on top of homogenate and centrifuged for 2 hr at 20,000 *g* at 4°C. The tube was then frozen at −80°C and cut at the 1,000 µl mark. The bottom piece of the centrifuge tube contained the non-LD fraction, which was allowed to thaw before being transferred to a new tube. The LD fraction was collected by cutting an ~4–6 mm piece from the top of the ice cylinder and placing it in a new 2 ml tube. To increase the purity of the LD fraction, this process was repeated once more. Briefly, 200 µl of 60% sucrose was added to the LD fraction. Next, 800 µl of lysis buffer was added and mixed followed by careful layering with 600 µl of lysis buffer and then centrifugation for 2 hr at 20,000 *g* at 4°C. After freezing at −80°C, the tube was cut and the LD fraction was collected by cutting an ~4–6 mm piece from the top of the ice cylinder and placing it in a new tube. Protein analysis was performed using the modified Lowry assay, as previously described (*Ferguson et al., 2017*), and Western blotting was performed as described below. MBOAT7 substrate (lysophosphatidylinositol) and product (phosphatidylinositol) lipids in the LD fraction were extracted and quantified using the targeted LC-MS/MS methods described below.

## Quantitation of phosphatidylinositol (PI) and lysophosphatidylinositol (LPI) lipids

A targeted lipidomic assay for LPI and PI lipids was developed using HPLC on-line electrospray ionization tandem mass spectrometry (LC/ESI/MS/MS). Plasma and tissues (liver, brain, pancreas) from mice fed chow or HFD with ASO injections for 20 weeks were analyzed for precise detection of LPI and PI species (*Figure 6*; *Figure 6—figure supplements 1–4*). Moreover, plasma LPI levels were measured over time after intraperitoneal injections of 18:0 and 18:1 LPI (*Figure 8—figure supplement 1*). *Standard Solutions:* The standards used in this assay were all purchased from Avanti Polar Lipids (LPI-16:0, LPI-18:0, LPI-18:1, LPI-20:4, PI-38:4). Internal standards used for these analyses were LPI-17:1, PI-34:1-d31; all of which were purchased from Avanti Polar Lipids. Standard LPI and PI species at concentrations of 0, 5, 20, 100, 500 and 2000 ng/ml were prepared in 90% methanol containing two internal standards at the concentration of 500 ng/ml. The volume of 5 µl was injected into the Shimadzu LCMS-8050 for generating the internal standard calibration curves. *HPLC Parameters:* A silica column (2.1 × 50 mm, Luna Silica, 5 µm, Phenomenex) was used for the separation of PI and LPI species. Mobile phases were A (water containing 10 mM ammonium acetate) and B (acetonitrile containing 10 mM ammonium acetate). Mobile phase B at 95% was used from 0 to 2 min at the flow rate of 0.3 ml/min and then a linear gradient from 95% B to 50% B from 2 to 8 min, kept at 50% B from 8 to 16 min, 50% B to 95% B from 16 to 16.1 min, kept 95% B from 16.1 to 24 min. *Mass Spectrometer Parameters:* The HPLC eluent was directly injected into a triple quadrupole mass spectrometer (Shimadzu LCMS-8050) and the analytes were ionized (ESI negative mode). Analytes were quantified using Selected Reaction Monitoring (SRM) and the SRM transitions (m/z) were 571 → 255 for LPI-16:0, 599 → 283 for LPI-18:0, 597 → 281 for LPI-18:1, 619 → 303 for LPI-20:4, 885 → 241 for PI-38:4, 583 → 267 for internal standard LPI-17:1, and 866 → 281 for internal standard PI-34:1-d31. *Data Analysis:* Software Labsolutions LCMS was used to get the peak area for both the internal standards and LPI and PI species. The internal standard calibration curves were used to calculate the concentration of LPI and PI species in the samples. All plasma LPI and PI species were normalized to the PI-34:1-d31 internal standard, while all tissue LPI species were normalized to the 17–1 LPI internal standard and all tissue PI species were normalized to the PI-34:1-d31 internal standard.

## Shotgun lipidomics to examine alterations in major phospholipid species

To more broadly examine the molecular lipid species in *Mboat7* knockdown livers we utilized a shotgun lipidomics method for semi-quantitation of multiple lipid species using a method we have previously described (*Gromovsky et al., 2018*). All the internal standards were purchased from Avanti

Polar Lipids, Inc (700 Industrial Park Drive, Alabaster, Alabama 35007, USA). Ten internal standards (12:0 diacylglycerol, 14:1 monoacylglycerol, 17:0 lysophosphatidylcholine, 17:0 phosphatidylcholine, 17:0 phosphatidic acid; 17:0 phosphatidylethanolamine, 17:0 phosphatidylglycerol, 17:0 sphingomyelin, 17:1 lysosphingomyelin, and 17:0 ceramide) were mixed together with the final concentration of 100 µM each. Total hepatic lipids were extracted using the method of Bligh and Dyer, with minor modifications. In brief, 50 µL of 100 µM internal standards were added to tissue homogenates and lipids were extracted by adding by adding MeOH/CHCl3 (v/v, 2/1) in the presence of dibutylhydroxytoluene (BHT) to limit oxidation. The $CHCl_3$ layer was collected and dried under $N_2$ flow. The dried lipid extract was dissolved in 1 ml the MeOH/CHCl3 (v/v, 2/1) containing 5 mM ammonium acetate for injection. The solution containing the lipid extract was pumped into the TripleTOF 5600 mass spectrometer (AB Sciex LLC, 500 Old Connecticut Path, Framingham, MA 01701, USA) at a flow rate of 40 µL/min for 2 min for each ionization mode. Lipid extracts were analyzed in both positive and negative ion modes for complete lipidome coverage using the TripleTOF 5600 System. Infusion MS/MSALL workflow experiments consisted of a TOF MS scan from m/z 200–1200 followed by a sequential acquisition of 1001 MS/MS spectra acquired from m/z 200 to 1200. The total time required to obtain a comprehensive profile of the lipidome was approximately 10 min per sample. The data was acquired with high resolution (>30000) and high mass accuracy (~5 ppm RMS). Data processing using LipidView Software identified 150–300 lipid species, covering diverse lipids classes including major glycerophospholipids and sphingolipids. The peak intensities for each identified lipid, across all samples were normalized against an internal standard from same lipid class for the semi-quantitation purpose.

## Identification and quantification of AA-Derived lipid mediators by LC-MS/MS

In order to quantitate the abundance of lipid mediators generated from arachidonic acid, livers collected from C57BL/6 mice exposed to each treatment condition were subjected to solid phase extraction (SPE) and targeted liquid chromatography-tandem mass spectrometry (LC-MS/MS) as previously described in detail (*Dalli et al., 2018*) Briefly, tissue was minced in ice-cold methanol containing internal deuterated standards ($d_4$-LTB$_4$, $d_4$-PGE$_2$ and $d_5$-LXA$_4$) used to assess extraction recovery. Samples were loaded on C18 SPE cartridges and eluted methyl formate fractions were concentrated under a gentle stream of $N_2$ gas. Samples were then resuspended in methanol:water (50:50) and analyzed using a high performance liquid chromatograph (HPLC, Shimadzu) coupled to a QTrap5500 mass spectrometer (AB Sciex) operating in negative ionization mode. Lipid mediators were identified using specific multiple reaction monitoring transitions, retention time, and diagnostic fragmentation spectra as compared with authentic standards. Quantification of lipid mediators was accomplished using calibration curves constructed with external standards for each mediator.

### RNA sequencing and analysis

RNA was isolated via RNAeasy lipid tissue mini kit (Qiagen) from livers in which mice were fed a high-fat diet and treated with ASOs for 20 weeks. RNA samples were checked for quality and quantity using the Bio-analyzer (Agilent). RNA-SEQ libraries were generated using the Illumina mRNA TruSEQ Directional library kit and sequenced using an Illumina HiSEQ4000 (both according to the Manufacturer's instructions). RNA sequencing was performed by the University of Chicago Genomics Facility. Raw sequence files will be deposited in the Sequence Read Archive before publication (SRA). Single-ended 50 bp reads were trimmed with Trim Galore (v.0.3.3, http://www.bioinformatics.babraham.ac.uk/projects/trim_galore) and controlled for quality with FastQC (v0.11.3, http://www.bioinformatics.bbsrc.ac.uk/projects/fastqc) before alignment to the *Mus musculus* genome (Mm10 using UCSC transcript annotations downloaded July 2016). Reads were aligned using STAR in single-pass mode (v.2.5.2a_modified, https://github.com/alexdobin/STAR) (*Dobin et al., 2013*) with standard parameters but specifying '–alignIntronMax 100000 –quantMode GeneCounts'. Overall alignment ranged from 88–99% with 61–75% mapping uniquely. Transcripts with fewer than one mapped read per million (MMR) in all samples were filtered out before differential expression (DE) analysis. The filtering step removed 12,692/24,411 transcripts (52%). Raw counts were loaded into R (http://www.R-project.org/) (*R Development Core Team, 2015*) and edgeR (*Robinson et al., 2010*) was used to perform upper quantile, between-lane normalization, and DE analysis. Values generated

with the cpm function of edgeR, including library size normalization and log2 conversion, were used in figures. Heat maps were generated using pheatmap (*Kolde, 2015*). DAVID (v.6.8) (*Huang et al., 2009*) was used to identify enriched functional annotations in DE gene ID lists relative to the set of 'expressed' genes (defined as having a median count across samples > 1 read per million mapped). Functional annotation to gene ontology was also performed using Ingenuity-IPA software (Ingenuity Systems, Inc Redwood City, CA.) as previously described (*Thomas and Bonchev, 2010*).

## Glucose and insulin tolerance tests

Glucose and insulin tolerance tests were conducted as previously described (*Warrier et al., 2015*; *Brown et al., 2010*; *Brown et al., 2008a*; *Brown et al., 2008b*; *Izem and Morton, 2009*; *Helsley et al., 2016*) in male mice following 20 weeks of concurrent chow and HFD-feeding with ASO treatment. During the glucose tolerance test plasma insulin and C-peptide levels were measured via ELISA following manufacturer's instructions (Millipore).

## Plasma lipid and lipoprotein analyses

Total plasma triacylglycerol levels (L-Type TG M, Wako Diagnostics) and total plasma cholesterol levels were quantified enzymatically (Infinity Cholesterol Reagent, Thermo/Fisher). The distribution of cholesterol across lipoprotein classes was performed by fast-protein liquid chromatography (FPLC) using tandem superose-6 HR columns coupled with an online enzymatic cholesterol quantification as previously described (*Warrier et al., 2015*; *Brown et al., 2010*; *Brown et al., 2008a*; *Brown et al., 2008b*; *Izem and Morton, 2009*; *Helsley et al., 2016*).

## In vivo insulin signaling studies

WT mice were fed a HFD for 20 weeks while injected with control or *Mboat7* targeted ASOs. At 20 weeks, mice were fasted overnight (7 p.m. to 7 a.m.) and were injected with saline or 0.35 U/kg insulin into the portal vein for 5 min, as previously described (*Lord et al., 2016*). After 5 min, mice were euthanized and tissues were collected for protein isolation and western blot analysis.

## Measurement of hepatic Very Low Density Lipoprotein (VLDL) Triglyceride Secretion in vivo

Control and *Mboat7* ASO-treated mice were fasted overnight, and the following morning rec were injected with 500 mg/kg Triton WR-1339 (tyloxapol, Sigma). and blood was collected every 90 min thereafter. To measure hepatic VLDL secretion, serum TG levels were measured using enzymatic biochemical assays at each time point as described below.

## General biochemical plasma and hepatic lipid analysis

Extraction of liver lipids and quantification of total plasma and hepatic triglycerides, cholesterol, and cholesterol esters was conducted using enzymatic assays as described previously (*Warrier et al., 2015*; *Brown et al., 2010*; *Brown et al., 2008a*; *Brown et al., 2008b*; *Izem and Morton, 2009*).

## Lysophosphatidylinositol acyltransferase biochemical assay

[1-$^{14}$C]-arachidonyl-CoA was obtained from American Radiolabeled Chemicals. Lysophosphatidylinositol (LPI) substrates (16:0 LPI, 18:0 LPI, and 18:1 LPI) and lipid standards used in the enzyme assays were obtained from Avanti Polar Lipids. Liver microsomes isolated from *Mboat7* ASO and control ASO treated mice on both chow and high fat diet was used to measure LPIAT activity. The assay buffer contained 50 mM Tris-HCL (pH 8.0), 150 mM NaCl, 50 µM 18:0-LPI, 20 µM [1-$^{14}$C]arachidonyl-CoA (0.025 µCi), and 15 µg of the microsomes in a total volume of 100 µL. Substrate was prepared in CHAPS (0.01 mM final concentration). The assay mixture was incubated for 30 min at 37°C, and the reaction was stopped by the addition of 1:2:1 (v/v/v) chloroform:methanol:2% orthophosphoric acid. The lipids were extracted and separated on a silica–TLC plate using chloroform/acetone/acetic acid/methanol/water (50:20:15:10:5, v/v) as the solvent system. The individual lipid molecules were identified by migration with respect to standards. Enzymatic products were monitored by autoradiogram, corresponding spots were scraped from the TLC plate, and the radioactivity was quantified with a liquid scintillation counter.

## Immunoblotting

Whole tissue homogenates were made from tissues in a modified RIPA buffer as previously described (*Warrier et al., 2015*; *Brown et al., 2010*; *Brown et al., 2008a*; *Brown et al., 2008b*; *Izem and Morton, 2009*), and protein was quantified using the bicinchoninic (BCA) assay (Pierce). Proteins were separated by 4–12% SDS-PAGE, transferred to polyvinylidene difluoride (PVDF) membranes, and proteins were detected after incubation with specific antibodies as previously described (*Warrier et al., 2015*; *Brown et al., 2010*; *Brown et al., 2008a*; *Brown et al., 2008b*; *Izem and Morton, 2009*). Quantification of blots was performed using ImageJ software (National Institute of Health). Information of antibodies is provided in the key resource table.

## Real-Time PCR analysis of gene expression

Tissue RNA extraction was performed as previously described for all mRNA analyses (*Warrier et al., 2015*; *Brown et al., 2010*; *Brown et al., 2008a*; *Brown et al., 2008b*; *Izem and Morton, 2009*). qPCR analyses were conducted as previously described (*Warrier et al., 2015*; *Brown et al., 2010*; *Brown et al., 2008a*; *Brown et al., 2008b*), and mRNA expression levels were calculated based on the ΔΔ-CT method. qPCR was conducted using the Applied Biosystems 7500 Real-Time PCR System. Primers used for qPCR are listed in the Key Resource Table.

## Histological analysis

Hematoxylin and eosin (H and E) staining of paraffin-embedded liver sections was performed as previously described (*Warrier et al., 2015*; *Brown et al., 2010*; *Brown et al., 2008a*; *Brown et al., 2008b*; *Izem and Morton, 2009*). Histopathologic evaluation was scored in a blinded fashion by a board-certified pathologist with expertise in gastrointestinal/liver pathology (Daniela S. Allende – Cleveland Clinic).

## Quantification of hepatic immune cell populations by flow cytometry

After 20 weeks of HFD-feeding and ASO injections, livers were excised, washed with 1X PBS, and immediately placed into RPMI with Type IV Collagenase (Sigma Aldrich, St. Louis, Missouri, Lot# 087K8630) and DNase I (Roche, Mannheim, German) for 45 min at 37° C. Digested clumps of liver were pressed through a 70 um strainer and washed with RPMI with 10% FBS. Cells were centrifuged at 50 g for 10 min; supernatant was then centrifuged at 50 g for 7 min. To pellet the NPC fraction, cells were centrifuged at 300 g for 7 min. Cells were resuspended in BD Pharm Lyse (BD Biosciences, San Jose, California) for 5 min on ice. Cells were washed with RPMI with 10% FBS and centrifuged at 300 g for 10 min. (Gibbons MA, American Journal of Respiratory and Critical Care Medicine, 184, 2011). For flow cytometry, cells were resuspended in FACS buffer (1x PBS, 1% BSA, 0.05% sodium azide). Cells were aliquoted into 96 well plates at a concentration of $\sim 1 \times 10^6$ cells/mL. Cells were centrifuged at 830 x g for 4 min, resuspended in 50 µl FACS buffer containing 0.5 ug of Fcγ Block (clone 93, eBioscience, San Diego, California), and incubated for 15 min at room temperature. After blocking, cells were stained with a fluorochrome-conjugated antibody panel CD206, Ly6c, CD3, CD4, CD8, CD11b, and CD11c. – all described in the Key Resource Table) for 30 min at 4° C in the dark. Cells were washed and centrifuged at 830 x g for 4 min twice with FACS buffer. Stained cells are resuspended in 200 µl of 1% paraformaldehyde and kept in the dark at 4°C overnight. Stained cells were centrifuged at 830 x g for 5 min. Stained cells were resuspended in 300 µl of FACS buffer, and data were collected on a LSRII flow cytometer (Becton Dickinson Immunocytometry systems, Mountain View, CA). Data collected on the LSRII were analyzed using FlowJo software (Tree Star, Inc, Ashland Oregon).

## Cell culture and generation of *MBOAT7-Deficient Hepatoma Cells*

Mycoplasma-tested hepatocellular carcinoma cells (Huh7) were cultured under standard conditions in Dulbecco-modified Eagle's minimum essential medium (D-MEM) (GIBCO, Life Technologies, Carlsbad, CA) supplemented with 10% fetal bovine serum (FBS, GIBCO), 1% l-glutamine, 1% penicillin-streptomycin and 1% nonessential amino acids in a 5% CO2-humidified chamber at 37°C. CRISPR-Cas9 genome editing was accomplished using methods previously described (*Ran et al., 2013*). MBOAT7 sgRNAs were designed by an online tool (https://www.benchling.com/) and cloned into the Lenti-CRISPER v2 vector (Addgene (*Ran et al., 2013*) with D10A nickase version of Cas9

(Cas9n)). Primers used in this study are listed in Supplementary Table X. *MBOAT7 KO* cell lines were generated following lentiviral transduction of the Lenti-CRISPER v2-Cas9 D10A- MBOAT7 sgRNA in Huh7 cells. *MBOAT7 KO* single cell clones were isolated and expanded following FACS sorting. *MBOAT7 KO* cells were validated by analyzing the expression of MBOAT7 by real-time PCR and Western blot. Primers used for gene editing are shown in the key resource table with the following primer names: *MBOAT7*-E5-Nick-5F, *MBOAT7*-E5-Nick-5R, *MBOAT7*-E5-Nick-3F, *MBOAT7*-E5-Nick-3R.

## Measurement of fatty acid oxidation rates in wild type and *MBOAT7Δ* Huh7 cells

Complete fatty acid oxidation to $CO_2$ was measured using methods we have previously described (*Brown et al., 2003*). In brief control and *MBOAT7Δ* HUH7 cells were grown to a confluency of 70–80% in DMEM. Upon reaching confluence, one million cells were seeded for both WT-HUH7 and *MBOAT7Δ* (n = 3) for 24 hr. After 24 hr the medium was replaced with fresh serum free DMEM medium with 0.4 μci of [1-$^{14}$C]-palmitic acid in 0.3% BSA/100 μM cold palmitate to each plate, and the plate was quickly placed in an airtight $CO_2$ collection chamber (60 ml Nalgene jar with a fitted rubber stopper, and hanging center-well collection bucket containing What-man filter paper soaked with 50 μl benzothonium hydroxide). Substrate incubation was carried out with different time points (0 min, 30 min, 60 min and 120 min) at 37°C. Following each incubation, the reaction was terminated by the addition of 100 μL of 0.5 M $H_2SO_4$ to the cells using syringe injection. Liberation of [$^{14}$C]-$CO_2$ was allowed to proceed for an additional 30 min, and then the center-well collection bucket was cut out of the collection chamber and delivered to a liquid scintillation vial. Production of [$^{14}$C]-$CO_2$ from [$^{14}$C]-palmitic acid was determined by liquid scintillation counter. Radiation count was normalized to amount of protein, as quantified by BCA assay (Pierce).

## Measurement of triacylglycerol and cholesterol ester hydrolysis rates in cells

Measurement of neutral lipid mobilization was accomplished using a pulse-chase approach essentially as described previously (*Brasaemle et al., 2000*). Both control and *MBOAT7Δ* Huh7 cells were grown till 80% confluent, one million cells were seeded in 6-well plates (WT-Huh7 and MBOAT7 KO, the next day fresh DMEM medium was added and the cells were incubated with 100 μM cold oleate complexed to bovine serum albumin + 1 μCi [$^3$H]-cholesterol + 0.5 μCi [$^{14}$C]-oleate for 24 hr. After 24 hr of pulse labeling-cells to reach steady state, cells were chased for different time points (0, 1 hr, 2 hr, 4 hr, 8 hr, 24, 48 hr) in media containing 6 μM triacsin C (from 1 mg/ml stock in Me$_2$SO) without supplemental fatty acids to examine turnover without the re-esterification (triacsin C blocks acyl-CoA synthetase activity). Cells were rinsed with phosphate buffered saline and harvested by scraping at various times; lipids were extracted as previously described and separated by thin layer chromatography using hexane:diethyl ether:acetic acid (80:20:1) as a solvent system. Triacylglycerol (TAG) and cholesterol ester spots were scraped off the plate, and the incorporation of radioactivity into TAG was determined by liquid scintillation counter. Radiation count was normalized to amount of protein, as quantified by BCA assay (Pierce).

## Measurement of de novo Lipogenesis in Cells

Measurement of de novo lipogenesis rates was accomplished by tracing [$^{14}$C]-acetate into [$^{14}$C]-triacylglycerol essentially as described previously (*Brasaemle et al., 2000*). Confluent cells of both wild type and *MBOAT7Δ* Huh7 cells were maintained in DMEM medium, from these 1 million cells of both WT and *MBOAT7Δ* were seeded to 6-well plates for 12 hr, after 12 hr the medium was replaced by serum free DMEM. To measure triacylglycerol synthesis, without the complication of parallel triacylglycerol hydrolysis, cells were treated with lipase inhibitors of triacylglycerol and cholesterol ester such as diethyl-p-nitrophenyl phosphate (E-600, 500 μM) and diethylumelliferyl phosphate (DEUP, 0.35 μM) for 1 hr. After 1 hr of lipase inhibitor treatment, the cells were incubated with human recombinant insulin (100 nM) as a stimulator for de novo lipogenesis for 30 min. After 30 min of insulin stimulation, the cells were incubated with 1 μCi [$^{14}$C]acetate and cells were harvested at various time points (30 min, 60 min, 120 min and 240 min) post substrate addition. From each time point cells were rinsed with PBS (twice), lipids were extracted as described above and

separated by thin layer chromatography using using hexane:diethyl ether:acetic acid (80:20:1) as a solvent system. Triacylglycerol (TAG) spot was scraped off of the plate, and the incorporation of [$^{14}$C]-acetate into [$^{14}$C]-triacylglycerol was determined by liquid scintillation counter. Radiation count was normalized to amount of protein, as quantified by BCA assay (Pierce).

## Fluorescence microscopy

In preparation for fluorescence microscopy, $5 \times 10^4$ cells of WT and *MBOAT7Δ* were plated onto 22 mm square glass coverslips in 4-well cell culture chamber slide containing growth media supplemented with 400 μM oleic acid. After 24 hr, medium was removed, washed with PBS (twice) and cells were fixed in 4% paraformaldehyde for 30 min. After 30 min of incubation, fixed cells were washed two times with PBS and then BODIPY (1 mg/mL) was added incubated for additional 30 min in dark. After incubation the cells were washed with PBS (twice) followed by mounting the slides with Pro-Long Gold antifade reagent with DAPI (4′,6- Diamidine-2′-phenylindole dihydrochloride). Images were acquired using a Leica DMIRB upright microscope (Leica Microsystems, GmbH, Wetzlar, Germany) equipped with a Retiga SRV camera and QCapture Pro software (QImaging, Surrey, BC, Canada).

## Western blot analysis

Microsome were isolated from single clones of *MBOAT7 Δ* and WT-HUH7 cells using microsome isolation kit from Abcam. Proteins were quantified from isolated microsome using BCA method, 30 μg of protein were loaded on SDA-PAGE gels, the expression of MBOAT7 is confirmed by using MBOAT7-rat primary antibody (1:1000) and anti-rat secondary antibody (1:5000).

## Statistical analysis

All experiments consisted of a minimum of three replicates and data are presented as mean ± SD. GraphPad Prism 8.1.1 was used for data analysis. Two-way ANOVAs with Bonferroni's multiple comparison tests were used to determine significant differences (p-value<0.0002(***),<0.0001(****)).

### Statistical analysis

Human data analyses were performed using R 3.1.0 (Vienna, Austria) and p<0.05 was considered statistically significant. All mouse data were analyzed using either one-way or two-way analysis of variance (ANOVA) where appropriate, followed by either a Tukey's or Student's t tests for post hoc analysis. Differences were considered significant at p<0.05. All mouse data analyses were performed using Graphpad Prism 6 (La Jolla, CA) software.

## Acknowledgements

This work is dedicated in loving memory to the late Dr. Lawrence 'Larry' Rudel, who passed away during the revision of this work (deceased, August 29[th], 2019). Larry's passion for science inspired many in the field of lipid and lipoprotein metabolism. We sincerely thank Dr. Hiroyuki Arai (University of Tokyo) for generously providing the MBOAT7 antibody used in this work. This work was supported by National Institutes of Health grants R01 HL122283 (JMB), R01 DK120679 (JMB), P50 AA024333 (JMB, DSA), P01 HL49373 (LLR), P01 HL30568 (AJL), U01 DK061732-15 (DSA), U01 AA021893-05 (DSA), P01 HL029582 (PLF), R00HL12172 (MC), R01 DK103637 (DSA), R01 HL106173 (MS), P01 GM095467 (MS), F32 HL136044 (BES), R21 AR067477 (JPK), and the American Heart Association (Postdoctoral Fellowships 17POST3285000 to RNH,15POST2535000 to RCS, and 19POST34380725 to IR). Development of lipid mass spectrometry methods reported here were supported by generous pilot grants from the Clinical and Translational Science Collaborative of Cleveland (4UL1TR000439) from the National Center for Advancing Translational Sciences (NCATS) component of NIH and the NIH Roadmap for Medical Research, the Case Comprehensive Cancer Center (P30 CA043703), the VeloSano Foundation, and a Cleveland Clinic Research Center of Excellence Award.

## Additional information

### Competing interests

Richard G Lee, Rosanne M Crooke, Mark J Graham: employee at Ionis Pharmaceuticals, Inc. The other authors declare that no competing interests exist.

### Funding

| Funder | Grant reference number | Author |
| --- | --- | --- |
| National Heart, Lung, and Blood Institute | R01-HL122283 | Jonathan Mark Brown |
| National Institute of Diabetes and Digestive and Kidney Diseases | R01-DK120679 | Jonathan Mark Brown |
| National Institute on Alcohol Abuse and Alcoholism | P50-AA-024333 | Daniela S Allende Jonathan Mark Brown |
| National Heart, Lung, and Blood Institute | P01 HL029582 | Paul L Fox |
| National Institute on Alcohol Abuse and Alcoholism | U01-AA021893 | Daniela S Allende |
| National Institute of Diabetes and Digestive and Kidney Diseases | U01-DK061732 | Daniela S Allende |
| National Institute of Diabetes and Digestive and Kidney Diseases | R01-DK103637 | Daniela S Allende |
| National Heart, Lung, and Blood Institute | P01-HL49373 | Lawrence L Rudel |
| National Heart, Lung, and Blood Institute | P01-HL30568 | Aldons J Lusis |
| National Heart, Lung, and Blood Institute | R00-HL12172 | Mete Civelek |
| National Heart, Lung, and Blood Institute | R01-HL106173 | Matthew Spite |
| National Heart, Lung, and Blood Institute | F32-HL136044 | Brian E Sansbury |

All coauthors are responsible for the content of this work, and different aspects of this work was funded by the National Institutes of Health (NIH) and the American Heart Association (AHA).

### Author contributions

Robert N Helsley, Data curation, Formal analysis, Validation, Investigation, Methodology, Writing—original draft, Writing—review and editing; Venkateshwari Varadharajan, Amanda L Brown, Anthony D Gromovsky, Rebecca C Schugar, Rakhee Banerjee, Data curation, Formal analysis, Investigation, Methodology, Writing—original draft, Writing—review and editing; Iyappan Ramachandiran, Kevin Fung, Investigation, Methodology; Mohammad Nasser Kabbany, Chelsea Finney, Preeti Pathak, Danny Orabi, Lucas J Osborn, William Massey, Renliang Zhang, Investigation, Methodology, Writing—original draft, Writing—review and editing; Chase K Neumann, Data curation, Investigation, Methodology, Writing—original draft, Writing—review and editing; Anagha Kadam, Investigation; Brian E Sansbury, Methodology, Writing—original draft, Writing—review and editing; Calvin Pan, Daniela S Allende, Mete Civelek, Aldons J Lusis, Data curation, Formal analysis, Methodology, Writing—original draft, Writing—review and editing; Jessica Sacks, Investigation, Writing—original draft, Writing—review and editing; Richard G Lee, Rosanne M Crooke, Mark J Graham, Resources, Methodology, Writing—original draft, Writing—review and editing; Madeleine E Lemieux, Data curation, Formal analysis; Valentin Gogonea, Data curation, Formal analysis,

Methodology, Writing—review and editing; John P Kirwan, Data curation, Methodology, Writing—original draft; Paul L Fox, Supervision, Investigation, Methodology; Lawrence L Rudel, Data curation, Formal analysis, Investigation, Methodology, Writing—original draft; Matthew Spite, Formal analysis, Investigation, Methodology, Writing—original draft, Writing—review and editing; J Mark Brown, Conceptualization, Resources, Data curation, Formal analysis, Supervision, Funding acquisition, Validation, Investigation, Visualization, Methodology, Writing—original draft, Project administration, Writing—review and editing

### Author ORCIDs
Robert N Helsley https://orcid.org/0000-0001-5000-3187
J Mark Brown https://orcid.org/0000-0003-2708-7487

### Ethics

Human subjects: Human MBOAT7 Expression Levels in Lean and Obese Subjects The majority of subjects recruited to examine MBOAT7 expression levels were morbidly obese bariatric surgery patients, but we were able to obtain liver biopsies from 10 subjects with a BMI under 30 as normal weight controls. For recruitment, adult patients undergoing gastric bypass surgery at Wake Forest School of Medicine were consented via written consent and enrolled by a member of the study staff following institutionally approved IRB protocols as previously described[41]. Exclusion criteria included: positive hepatitis C antibody, positive hepatitis B surface antigen, history of liver disease other than NAFLD, Childs A, B, or C cirrhosis, past or present diagnosis/treatment of malignancy other than non-melanocytic skin cancer, INR greater than 1.8 at baseline or need for chronic anticoagulation with warfarin or heparin products, use of immunomodulation for or history of inflammatory diseases including but not limited to malignancy, rheumatoid arthritis, psoriasis, lupus, sarcoidosis and inflammatory bowel disease, and greater or equal to 7 alcohol drinks per week or 3 alcoholic drinks in a given day each week. In addition to bariatric surgery patients, a small number of non-obese subjects (body mass index <30.0) consented to liver biopsy during elective gall bladder removal surgery (n=10). Each subject was assigned a unique identifier which was used throughout the study and did not include any identifiable information about the patient such as name, address, telephone number, social security number, medical record number or any of the identifiers outlined in the HIPAA Privacy Rule regulations. Only the principal investigator had access to the code linking the unique identifier to the study subject. Basic clinical information was obtained via self-reporting and a 15 ml baseline blood sample was obtained at the time of enrollment. A subset of this cohort has been previously described[20,41]. At the time of surgery, the surgeon collected a roughly 1-gram sample from the lateral left lobe. Wedge biopsies were rinsed with saline and immediately snap frozen in liquid nitrogen in the operating room before subsequent storage at -80°C. For data shown in Figure 7 showing levels of MBOAT7 substrate and product lipids, de-identified patient samples from the Cleveland Clinic hepatology clinic (IRB # 10-947) were analyzed. These patients had biopsy proven Ishak fibrosis scores[42] of 0 (normal) or 4 (advanced fibrosis). For analysis of hepatic MBOAT7 expression, RNA isolated from liver biopsies were used for quantitative real time PCR (qPCR) as described below.

Animal experimentation: Rat Studies of Diet-Induced Obesity Sprague Dawley Rats were received at 12 weeks of age and were housed in individual cages, kept at a constant temperature and ambient humidity in a 12-h light/dark cycle. Animals were then randomly assigned to either a standard chow diet or a high-fat diet (D12492, 60% fat, Research Diets, New Brunswick, NJ, USA) ad libitum to establish diet-induced obesity as previously described[20]. After 6 months of HFD-feeding, livers were excised for standard qPCR analysis of Mboat7 expression. Hybrid Mouse Diversity Panel 92 inbred strains of 8-week-old male mice (180 individual mice) were fed a high fat, high sucrose diet (D12266B, Research Diets, New Brunswick, NJ) for 8 weeks before tissue collection[44]. Gene expression of Mboat7 in white adipose tissue and liver were measured and correlated with obesity related traits using biweight midcorrelation analysis as previously described[4]. Mouse Studies of Mboat7 Loss of Function To explore the role of Mboat7 in diet-induced obesity, NAFLD progression, and insulin resistance, we utilized an in vivo knockdown approach in 8-week old adult mice. Selective knockdown of Mboat7 was accomplished using 2'-O-ethyl (cET) modified antisense oligonucleotides (ASO). All ASOs used in this work were synthesized, screened, and purified as described previously[45] by Ionis Pharmaceuticals, Inc (Carlsbad, CA). For Mboat7 knockdown studies, adult (8 week

old) male C57BL/6 mice were purchased from Jackson Labs (Bar Harbor, ME USA), and maintained on either a standard rodent chow diet or a high fat diet (HFD, D12492 from Research Diets Inc) and injected intraperitoneally biweekly with 12.5 mg/kg of either non-targeting control ASO or one of two independent ASOs directed against murine Mboat7 for a period of 20 weeks. Similar results were seen with two independent ASOs targeting different regions of the Mboat7 mRNA, hence key data using one Mboat7 ASO are shown. All rodents were maintained in an Association for the Assessment and Accreditation of Laboratory Animal Care, International-approved animal facility, and all experimental protocols were approved by the Institutional Animal Care and use Committee of the Cleveland Clinic (IACUC protocols # 2015-1519 and # 2018-2053).

### Decision letter and Author response
Decision letter https://doi.org/10.7554/eLife.49882.036
Author response https://doi.org/10.7554/eLife.49882.037

## Additional files
### Supplementary files
• Transparent reporting form DOI: https://doi.org/10.7554/eLife.49882.028

### Data availability
Dataset are available through NCBI: Accession numbers GSE138945, GSE138946, and GSE138947.

The following datasets were generated:

| Author(s) | Year | Dataset title | Dataset URL | Database and Identifier |
|---|---|---|---|---|
| Robert N Helsley, Venkateshwari Varadharajan, Amanda L Brown, Anthony D Gromovsky, Rebecca C Schugar, Iyappan Ramachandiran, Kevin Fung, Mohammad Nasser Kabbany, Rakhee Banerjee, Chase Neumann, Chelsea Finney, Preeti Pathak, Danny Orabi, Lucas J Osborn, William Massey, Renliang Zhang, Anagha Kadam, Brian E Sansbury, Calvin Pan, Jessica Sacks, Richard G Lee, Rosanne M Crooke, Mark J Graham, Madeleine E Lemieux, Valentin Gogonea, John P Kirwan, Daniela S Allende, Mete Civelek, Paul L Fox, Lawrence L Rudel, Aldons J Lusis, Matthew Spite, J Mark Brown | 2019 | Obesity-linked suppression of MBOAT7 promotes liver injury | https://www.ncbi.nlm.nih.gov/geo/query/acc.cgi?acc=GSE138947 | NCBI Gene Expression Omnibus, GSE138947 |
| Robert N Helsley, Venkateshwari Varadharajan, Amanda L Brown, Anthony D Gromovsky, Rebec- | 2019 | MBOAT7's role in progression of Non-Alcoholic Fatty Liver Disease (NAFLD) and Non-Alcoholic Steatohepatitis (NASH) | https://www.ncbi.nlm.nih.gov/geo/query/acc.cgi?acc=GSE138945 | NCBI Gene Expression Omnibus, GSE138945 |

| | | | | |
|---|---|---|---|---|
| ca C Schugar, Iyappan Ramachandiran, Kevin Fung, Mohammad Nasser Kabbany, Rakhee Banerjee, Chase Neumann, Chelsea Finney, Preeti Pathak, Danny Orabi, Lucas J Osborn, William Massey, Renliang Zhang, Anagha Kadam, Brian E Sansbury, Calvin Pan, Jessica Sacks, Richard G Lee, Rosanne M Crooke, Mark J Graham, Madeleine E Lemieux, Valentin Gogonea, John P Kirwan, Daniela S Allende, Mete Civelek, Paul L Fox, Lawrence L Rudel, Aldons J Lusis, Matthew Spite, J Mark Brown | | | | |
| Robert N Helsley, Venkateshwari Varadharajan, Amanda L Brown, Anthony D Gromovsky, Rebecca C Schugar, Iyappan Ramachandiran, Kevin Fung, Mohammad Nasser Kabbany, Rakhee Banerjee, Chase Neumann, Chelsea Finney, Preeti Pathak, Danny Orabi, Lucas J Osborn, William Massey, Renliang Zhang, Anagha Kadam, Brian E Sansbury, Calvin Pan, Jessica Sacks, Richard G Lee, Rosanne M Crooke, Mark J Graham, Madeleine E Lemieux, Valentin Gogonea, John P Kirwan, Daniela S Allende, Mete Civelek, Paul L Fox, Lawrence L Rudel, Aldons J Lusis, Matthew Spite, J Mark Brown | 2019 | LPI's role in progression of Non-Alcoholic Fatty Liver Disease (NAFLD) and Non-Alcoholic Steatohepatitis (NASH) | https://www.ncbi.nlm.nih.gov/geo/query/acc.cgi?acc=GSE138946 | NCBI Gene Expression Omnibus, GSE138946 |

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
