## [Decision Letter]

[Editors’ note: a previous version of this study was rejected after peer review, but the authors submitted for reconsideration. The first decision letter after peer review is shown below.]

Thank you for submitting your work entitled "Obesity-Linked Suppression of Membrane-Bound *O*-Acyltransferase 7 (*MBOAT7*<) Drives Non-Alcoholic Fatty Liver Disease" for consideration by *eLife*. Your article has been reviewed by three peer reviewers, one of whom is a member of our Board of Reviewing Editors, and the evaluation has been overseen by a Senior Editor. The reviewers have opted to remain anonymous.

Our decision has been reached after consultation between the reviewers. Based on these discussions and the individual reviews below, we regret to inform you that your work will not be considered further for publication in *eLife*. Although all reviewers believed that the study addressed an important problem, they also felt that the present data did not provide a sufficiently definitive mechanistic link between MBOAT7 and the development of hepatic pathology at this stage. Further, the extent of revisions required is beyond that expected for an *eLife* revision. However, the editors and reviewers are open to the possibility of evaluating a suitably revised manuscript as a de novo submission should you be able to substantially extend the current findings and respond to the reviewers concerns.

Reviewer #1:

Mboat7 is a lysoPI acyltransferase that has been identified as a candidate gene in GWAS studies for NAFLD. In this report, Helsley provide evidence that reduced Mboat7 activity drives the development of obesity-induced NAFLD in mouse models. They knocked down Mboat7 expression with an ASO in high-fat diet fed mice and found that Mboat7 deficient mice developed steatosis and inflammation in mouse liver, and insulin resistance. With respect to the mechanism involved, they showed that the levels of several LPIs and PIs were changed by Mboat7 knockdown. Finally, they showed that LPI treatment increased the expression of inflammatory genes in the livers of Mboat7-deficient mice, but not in control mice. Although the studies are of interest and obvious clinical relevance, the mechanistic insight into how Mboat7 affects hepatic lipid metabolism is limited at this stage.

Major points:

1) Using more than one Mboat7 ASO would increase the strength of the conclusions.

2) What is the mechanism causing lipid accumulation in the livers of ASO-treated mice? Is the expression of Srebp1c and lipogenic genes increased? The authors showed cholesterol in plasma lipoprotein particles was not altered, how about the TG profile in plasma lipoprotein particles? Is there any defect in VLDL secretion?

3) The diet regimen employed does not typically lead to fibrosis in mice. The authors showed increased expression of inflammatory and fibrotic genes, but to establish fibrosis they need to show Trichrome or Sirius staining. Similarly, they claim that stellate cells were activated, but need more direct data to support this.

4) The authors report that 16:0 and 18:1 LPI was increased upon Mboat7 suppression, but the levels of these two LPIs (~0.05 pmol/mg) are very low compared to 18:0 LPI (~5 pmol/mg). It is difficult to see how this change could be driving the phenotype. Is the total LPI level significantly increased? Is there any change in total PI levels? Why did the authors inject mice with 18:0 LPI, which was not altered by Mboat7 suppression, but not 16:0 or 18:1 LPI that was increased by Mboat7 knockdown? Do different LPI species have different effects?

5) For LPI treatment experiment, did the authors inject HFD fed mice or chow diet fed mice? Will an increase in LPI in chow diet fed mice induce a phenotype similar to HFD fed mice?

6) Does LPI treatment also affect other genes identified by RNA seq in figure 3? In other words, does LPI contribute to other phenotypes including steatosis and insulin resistance?

Reviewer #2:

Major comments::

The main focus of this paper is that reduced expression of MBOAT7 in the liver drives NAFLD (per the title) and NAFLD consequences. In my view, it's a quite interesting paper but is fundamentally flawed, as the ASO targeting MBOAT7 in murine liver also reduces expression of TMC4, the other candidate gene. Given this, it's not possible to determine which effects are due to MBOAT7 depletion, which is the focus of the paper.

Figure 1A-D; need more controls (positive and negative). Units for x axes in D-F are not clear. Also, In the Discussion section the authors state a major finding is that hepatic expression of MBOAT7 is reduced in obese human and rodents independent of rs641738 status, so would be nice to see this data.

Figure 2A; pertinent negative controls should be shown. The ASO used targets liver, adipose and reticuloendothelial system. Figure 2A and Figure 2—figure supplement 1 only show mRNA data for liver, brain, spleen, heart, skeletal muscle. It would be good to include data in adipose. All this should be in main figures. Also, given the main thrust of the study is NAFLD/NASH, the authors should show collagen staining and fibrosis pathology scores.

Figure 3E presents a major problem as the MBOAT7 ASO lowers mRNA for TMC4, making the rest of the experiments (most of the rest of the paper) not possible to interpret for MBOAT7 depletion. If the point of the paper is that both genes are coordinately regulated and somehow both function in these phenotypes, then more work needs to be done to understand this. However, the current paper is about MBOAT7.

Figure 5; results for other phospholipids and lysophospholipids should be shown. Is this defect specific for PI/LPI?

Figure 6; it's not clear exactly how LPI is given, how much, and whether it is "physiological".

Reviewer #3:

This is the first characterization of MBOAT7 knockdown mice. The characterization is good and complete with interesting findings. There are several important weaknesses- they don't actually establish whether MBOAT7 or TMC4 are independently or cooperatively involved in the resulting phenotype. I don't think with the experiments presented they have firmly established with the alterations in lipid species found are responsible for the inflammation and stellate cell activation. It is important they address my third major point prior to any publication as it has important implication if true. Finally, they don't address why TGs accumulate in liver. The latter seem pretty basic given this is how the region was identified in the first place. Overall, I think this is marginal for acceptance, but certainly more experiments would be required.

Major points:

1) The effect of the ASO on FAT Mboat7 expression should be included.

2) Regarding the experimental design of Figure 6. It is not clear whether the amount of 18:0 LPI injected resulted in a 2-fold increase in the total LPI concentration or in the amount of 18:0. The later would be appropriate for these experiments. It seems to directly link the LPI and the function of MBOAT7, only the species that were altered should induce the inflammation and stellate cell activation. There is a lack of controls in this important experiment, which should include species that were not altered by MBOAT7 knockdown.

3) What is the mechanism by which the MBOAT7 knockdown results in TG, CE and FC accumulation in liver?

[Editors’ note: what now follows is the decision letter after the authors submitted for further consideration.]

Thank you for submitting your article "Obesity-linked suppression of membrane-bound *O*-Acyltransferase 7 (*MBOAT7*) drives non-alcoholic fatty liver disease" for consideration by *eLife*. Your article has been reviewed by three peer reviewers, one of whom is a member of our Board of Reviewing Editors, and the evaluation has been overseen by Anna Akhmanova as the Senior Editor. The following individuals involved in review of your submission have agreed to reveal their identity: Dennis Vance (Reviewer #3).

The reviewers have discussed the reviews with one another and the Reviewing Editor has drafted this decision to help you prepare a revised submission.

Summary:

The reviewers and editors agree that the manuscript has been improved since the original submission and several issues have been clarified. However, the mechanism underlying the steatosis is still unclear. The new data regarding alterations to the liver lipid droplet proteome are interesting but do not establish whether these changes are a cause or a consequence of steatosis. Given that steatosis is the focus of the paper, the consensus is that this issue must be addressed prior to publication.

Essential revisions:

1) Direct evidence of how TG, CE and free cholesterol accumulates in Mboat7-deficient livers is not provided. A higher abundance of GPAT4 in the lipid droplet fraction was observed, but it is unclear where the acyl-CoA substrates for GPAT are coming from if there is no increase in de novo lipogenesis or uptake from circulation. The abundance of PLIN2, PLIN3, GPAT4, and CCTα on lipid droplets may merely reflect the presence of more lipid droplets. Increased abundance of GPAT4 and CCTα on lipid droplets does not explain why the livers of MBOAT7-deficient mice contain more CE and free cholesterol. mRNAs linked to lipogenesis were not measured in conditions that stimulate lipogenesis (i.e. re-feeding) but were instead measured after fasting or at the start of the light cycle.

The following experiments could be performed in the given time frame to address this central issue:

- Measure lipogenic mRNAs in re-fed conditions (see papers by Horton and colleagues on SREBP1-dependent lipogenesis).

- The rate of liver lipid synthesis can be assessed by i.p injection of radiolabeled glycerol and measurement of glycerol incorporation into lipid classes after separation by TLC as in Miyazaki et al., 2001 or Millar et al., 2006.

- Alternatively, if they believe their data is consistent with an impairment in TG mobilization from LDs, then the rate of liver LD lipolysis can be assessed as in Schweiger et al., 2014. These measurements could be done in primary hepatocytes/cell culture models if that is easier to achieve.

- de novo lipogenesis as a contributor to fat accumulation could be assessed by injecting mice with 14C-acetate (after they had been fasted O/N and re-fed for 6 hours) as in Lian et al., 2018.

2) In the experiment in which LPI species were administered to mice, only mRNAs related to inflammation were assessed. Analysis of hepatic lipid metabolism should be included in this experiment, especially since the focus of the manuscript has shifted from fibrosis/inflammation to hepatic steatosis.

---

## [Author Response]

[Editors’ note: the author responses to the first round of peer review follow.]

Reviewer #1:Mboat7 is a lysoPI acyltransferase that has been identified as a candidate gene in GWAS studies for NAFLD. In this report, Helsley provide evidence that reduced Mboat7 activity drives the development of obesity-induced NAFLD in mouse models. They knocked down Mboat7 expression with an ASO in high-fat diet fed mice and found that Mboat7 deficient mice developed steatosis and inflammation in mouse liver, and insulin resistance. With respect to the mechanism involved, they showed that the levels of several LPIs and PIs were changed by Mboat7 knockdown. Finally, they showed that LPI treatment increased the expression of inflammatory genes in the livers of Mboat7-deficient mice, but not in control mice. Although the studies are of interest and obvious clinical relevance, the mechanistic insight into how Mboat7 affects hepatic lipid metabolism is limited at this stage.Major points:1) Using more than one Mboat7 ASO would increase the strength of the conclusions.

Given this valid concern we have now used two independent ASOs targeting different regions of the Mboat7 messenger RNA, and see consistent effects on hepatic lipid accumulation. We have added a new supplemental figure (Figure 2—figure supplement 4) in response to this concern.

2) What is the mechanism causing lipid accumulation in the livers of ASO-treated mice? Is the expression of Srebp1c and lipogenic genes increased? The authors showed cholesterol in plasma lipoprotein particles was not altered, how about the TG profile in plasma lipoprotein particles? Is there any defect in VLDL secretion?

To address this, we have completed a number of additional experiments to understand the mechanism by which MBOAT7 loss of function promotes hepatic steatosis. First, we have repeatedly seen that Mboat7 knockdown does not significantly alter Srebp1c or lipogenic gene expression. We have also consistently seen no alterations in steady state plasma triglyceride levels or VLDL secretion rates. These new data for lipogenic gene expression and VLDL secretion are now included in a new main figure (Figure 9). After ruling out these common causes of hepatic steatosis, we instead focused our attention on whether MBOAT7 knockdown could drive steatosis by locally altering the lipid and proteome composition of cytosolic lipid droplets (i.e. impacting lipid storage at the lipid droplet interface). The rationale for pursuing this possibility was that a recent manuscript (Mancina et al., 2016) suggested that MBOAT7 can co-purify with cytosolic lipid droplets in subcellular fractionation experiments. Using sucrose gradient lipid droplet fractionation protocols, we have now confirmed that MBOAT7 can be found on cytosolic lipid droplets in control mice but is essentially absent on lipid droplets from Mboat7 ASO-treated mice (Figure 9F). In parallel with MBOAT7 localization to lipid droplets, we found the lipid droplets isolated from Mboat7 ASO-treated mice show striking increases in MBOAT7 substrates (18:0 LPI and 20:4 LPI) and reduction in MBOAT7 products (38:4 PI). In fact, the differences seen in LPI and PI lipids are much more apparent in isolated lipid droplets (Figure 9) than what is seen in whole liver homogenates (Figure 6). Aligned with alterations in LPI and PI lipids, lipid droplets isolated from Mboat7 knockdown livers had a clear rearrangement of lipid droplet-associated proteins. Lipid droplets isolated from Mboat7 ASO-treated livers have increased abundance of several lipid droplet-associated proteins including perilipin 2, perilipin 3, as well as the lipid synthesizing enzymes CTP:phosphocholine cytidylyltransferase α (CCTα) and glycerol-3-phosphate acyltransferase 4 (GPAT4) (Figure 9F). Aligned with these new findings we have added the following paragraph to the Discussion section of the revised manuscript:

“Steatosis Seen with Mboat7 Knockdown is Associated With Reorganization of the Hepatic Lipid Droplet Lipidome and Proteome. Mboat7 loss of function results in a striking accumulation of neutral lipids including triglycerides and cholesteryl esters (Figure 2e-2l), yet the mechanism(s) behind this are poorly understood. We therefore examined several potential mechanisms driving the mixed hepatic steatosis in Mboat7 ASO-treated mice. First, we examined the expression of genes involved in lipogenesis, fatty acid oxidation, and cholesterol sensing and export under both fed and fasted conditions (Figure 9). Unlike many other models of hepatic steatosis, there were no significant alterations in lipogenic gene expression either in the fed or fasted state with Mboat7 knockdown (Figure 9a). Unexpectedly, the expression of carnitine palmitoyl transferase 1 (Cpt1) was significantly elevated with Mboat7 knockdown (Figure 9a), but this would not be expected to promote the accumulation of neutral lipids. The expression of enzymes involved in cholesterol biosynthesis (Hmgcr and Hmcgs1) were modestly reduced in Mboat7 ASO-treated mice, but only in the fasted state (Figure 9a). In addition, expression of the cholesterol efflux regulator Abca1 was elevated in Mboat7 ASO-treated mice, but only in the fasted state. Altogether, these minor differences in hepatic gene expression are unlikely to be driving the lipid accumulation in Mboat7 ASO treated mice. Next, we evaluated whether Mboat7 may influence the export of neutral lipids via packaging on nascent very low density lipoproteins (VLDL). However, Mboat7 ASO-treated mice did not have significant differences in VLDL-TG secretion during a detergent block (Figure 9b). Another common cause of fatty liver is increased delivery of adipose-derived fatty acids to the liver, as is commonly seen with prolonged fasting and certain types of lipodystrophies. However, Mboat7 knockdown did not alter basal or catecholamine-stimulated adipocyte lipolysis in vivo (Figure 9c), ruling out a role for altered adipose lipolysis as a contributing factor. Finally, we examined the possibility that MBOAT7 may be a determinate of metabolism locally at the surface of cytosolic lipid droplets to regulate hepatic steatosis, given that one recent study reported that MBOAT7 can localize to cytosolic lipid droplets^2^. We confirmed that MBOAT7 can indeed be found in lipid droplets isolated by sucrose gradient fractionation (Figure 9f), and plays a regulatory role in both the lipidome and proteome of cytosolic lipid droplets. Cytosolic lipid droplets isolated from Mboat7 ASO-treated mice showed marked accumulation of 18:0 LPI and 20:4 LPI, but not 16:0 LPI or 18:1 LPI (Figure 9d). This is in stark contrast to what is seen in the whole liver, where Mboat7 knockdown instead promotes accumulation of 16:0 LPI and 18:1 LPI (Figure 6d). Also, lipid droplets isolated from Mboat7 ASO-treated mice have a reduced level of 38:4 PI but a reciprocal increase in 36:3 and 38:3 PI (Figure 9e), which is quite different when to compared to effects in whole liver (Figure 6e). In addition to alterations in the lipid droplet lipidome, Mboat7 knockdown is associated with accumulation of several proteins involved in lipid synthesis and storage on isolated lipid droplets (Figure 9f). Both perilipin 2 and 3 (PLIN2 and PLIN3) were much more abundant on lipid droplets isolated from Mboat7 ASO-treated mice, as were the critical lipid synthetic enzymes CTP:phosphcholine cytidylyltransferase α (CCTα) and glycerol-3-phosphate 4 (GPAT4). Given that recent reports have shown that lipid droplet targeting of CCTα and GPAT4 are critical regulators of the overall size and triglyceride storage capacity of lipid droplets^30-32^, these proteomic alterations at the lipid droplet surface are likely to contribute to the hepatic steatosis seen in Mboat7 ASO-treated mice.”

3) The diet regimen employed does not typically lead to fibrosis in mice. The authors showed increased expression of inflammatory and fibrotic genes, but to establish fibrosis they need to show Trichrome or Sirius staining. Similarly, they claim that stellate cells were activated, but need more direct data to support this.

This is a valid point that high fat diet feeding does not typically lead to measurable fibrosis in mice. Since our original submission we have done a number of experiments to test whether MBOAT7 knockdown indeed stimulates bridging fibrosis as is seen in advanced human liver disease, but as brought forth by this astute reviewer we failed to find pathological or biochemical support of fibrosis in this mouse model. First, we had a board-certified liver pathologist (Dr. Daniela Allende, Cleveland Clinic) blindly score H&E-, pirosirius Red, and trichrome-stained liver sections from HFD-fed control and Mboat7 ASO-treated mice, and in all cases there was no measureable fibrosis scoring above background. To indirectly quantitate fibrosis potential in a complimentary way we analyzed liver hydroxyproline content biochemically, and again found no indication that hydroxyproline levels were altered in Mboat7 ASO-treated mice. Finally, we performed some preliminary studies in primary stellate cells treating with MBOAT7 substrate lipids (LPIs) and did not see any obvious signs of pro-fibrotic responses in primary stellate cells. Given that we do not see any signs of even early stage fibrosis in Mboat7 ASO-treated mice, we have removed any discussion regarding effects on fibrosis and instead focus the discussion on the striking hepatic lipid accumulation seen with MBOAT7 loss of function in high fat fed mice. Although beyond the scope of the current study, we are now generating in parallel either hepatocyte, macrophage, or stellate cell specific Mboat7 knockout mice and will subject them to models of fibrosis induction (carbon tetrachloride, methionine/choline-deficient diet, bile duct ligation, etc.) to understand the cell autonomous effect of Mboat7 in pro-fibrotic responses.

To make this issue clear for the reader we have removed any claims that fibrosis was found in our studies, and have added the following sentences to the Discussion section:

“However, an important limitation of our studies is that high fat diet feeding in mice is not sufficient to drive bridging fibrosis. In future studies it will be important to examine how Mboat7 loss of function impacts the development of fibrosis in appropriate fibrosis-prone animal models, and whether Mboat7 expression in hepatic stellate cells plays a regulatory role in the progression from NASH to cirrhosis.”

4) The authors report that 16:0 and 18:1 LPI was increased upon Mboat7 suppression, but the levels of these two LPIs (~0.05 pmol/mg) are very low compared to 18:0 LPI (~5 pmol/mg). It is difficult to see how this change could be driving the phenotype. Is the total LPI level significantly increased? Is there any change in total PI levels? Why did the authors inject mice with 18:0 LPI, which was not altered by Mboat7 suppression, but not 16:0 or 18:1 LPI that was increased by Mboat7 knockdown? Do different LPI species have different effects?

This is a very insightful comment that was generally shared by all three reviewers, so we felt it was important to perform additional experiments to address the concern. We originally injected mice with 18:0 LPI because it by far is the most abundant LPI in the circulation, and we followed a similar logic as this astute reviewer that the most abundant species is the most likely culprit as a signaling lipid. We also knew from the literature that 18:0 LPI was previously described as a bona fide MBOAT7 substrate in biochemical assays from the Voelker laboratory (Gijón et al., 2008). We also now show that the acylation of 18:0 LPI is lower in Mboat7 ASO-treated mice (Figure 6A), particularly when challenged with a high fat diet. However, we agree that the primary changes seen with Mboat7 ASO treatment in the liver is instead the more minor species 16:0 and 18:1. To better understand how these more minor species of LPI (16:0 and 18:1) could impact liver disease in the context of Mboat7 loss of function, we performed new experiments examining whether direct injection of 16:0 or 18:1 LPI could induce liver injury in control or Mboat7 ASO-treated mice. Results from these new experiments are now included as a new main figure (Figure 8), and the results from the 18:0 LPI injections are included as an online supplemental figure to provide a comprehensive look at LPI subspecies effects (Figure 8—figure supplement 4). Interestingly, both 18:1 and 18:0 LPI induced expression of pro-inflammatory and fibrosis marker genes in a MBOAT7-dependent manner, but this was not apparent with 16:0 LPI administration. These new data provide additional evidence in support of substrate-specific effects in vivo, and highlight the importance of 18 carbon LPIs as potential proinflammatory and pro-fibrotic signaling lipids when MBOAT7 activity is compromised.

5) For LPI treatment experiment, did the authors inject HFD fed mice or chow diet fed mice? Will an increase in LPI in chow diet fed mice induce a phenotype similar to HFD fed mice?

To help get at this question, we have now performed LPI injection experiments in either chow-fed or HFD-fed cohorts of mice, and these new experiments are described in the revised manuscript. We have found that the ability of 18 carbon LPIs to induce pro-inflammatory and fibrosis related gene expression in Mboat7 ASO treated mice really only occurs when mice are fed a high fat diet and not seen in chow-fed mice. We have included a new supplementary figure to show the modest transcriptional changes following 18:0 LPI administration in chow-fed mice, but also show that some of these same genes are altered with Mboat7 knockdown and high fat diet feeding (Figure 8—figure supplement 2).

6) Does LPI treatment also affect other genes identified by RNA seq in figure 3? In other words, does LPI contribute to other phenotypes including steatosis and insulin resistance?

In parallel experiments described in question 5 above, we have injected LPI into control and Mboat7 ASO-treated mice maintained on a chow diet, and performed metabolic phenotyping including hepatic lipid quantification and unbiased RNA sequencing to address this question. In chow-fed mice we did not find that LPI injection promoted steatosis in either control or Mboat7 ASO-treated mice. We have included a new supplementary figure to show the modest transcriptional changes following 18:0 LPI administration in chow-fed mice, but also show that some of these same genes are altered with Mboat7 knockdown and high fat diet feeding (Figure 8—figure supplement 2).

Reviewer #2:Major comments:The main focus of this paper is that reduced expression of MBOAT7 in the liver drives NAFLD (per the title) and NAFLD consequences. In my view, it's a quite interesting paper but is fundamentally flawed, as the ASO targeting MBOAT7 in murine liver also reduces expression of TMC4, the other candidate gene. Given this, it's not possible to determine which effects are due to MBOAT7 depletion, which is the focus of the paper.

We have now clarified the role of Mboat7, but not Tmc4, in hepatic steatosis by generating Tmc4-/- mice. Importantly, global deletion of Tmc4 does not result in hepatic steatosis, but Mboat7 ASO treatment promotes striking steatosis when mice are fed a high fat diet (Figure 4).

Figure 1A-D need more controls (positive and negative). Units for x axes in D-F are not clear. Also, In the Discussion section the authors state a major finding is that hepatic expression of MBOAT7 is reduced in obese human and rodents independent of rs641738 status, so would be nice to see this data.

We apologize for this confusion. The units for panels D-F were relative quantitation of mRNA abundance from a previously performed microarray analysis (Affymetrix HT_MG-430A) on tissue isolated from the hybrid mouse diversity panel by the Lusis laboratory (Parks et al., 2015). As requested, we have now fixed these issues with this figure, and have included the rs641738 genotypes within each group as requested.

Figure 2A; pertinent negative controls should be shown. The ASO used targets liver, adipose and reticuloendothelial system. Figure 2A and Figure 2—figure supplement 1 only show mRNA data for liver, brain, spleen, heart, skeletal muscle. It would be good to include data in adipose. All this should be in main figures.

We have seen that our control ASO behaves very similar to a saline control group, and have now included a supplemental figure showing that two distinct ASO targeting different regions of the Mboat7 mRNA have similar effects on hepatic steatosis (Figure 2—figure supplement 4). To be more comprehensive in showing the reader where the ASOs target knockdown, we have now included figures showing the knockdown efficiency of Mboat7 ASO in liver (Figure 2), adipose (Figure 2—figure supplement 1), and other tissue including brain, spleen, heart, and skeletal muscle (Figure 2—figure supplement 2). It is important to note that despite significant knockdown of Mboat7 in Mboat7 ASO-treated mice, there are no differences in adiposity (Figure 2—figure supplement 1), basal and catecholamine-stimulated lipolysis (Figure 9c), energy expenditure (Figure 2—figure supplement 3), and only very minor differences in adipose tissue gene expression (Figure 2—figure supplement 1). Therefore, our data do not support a strong role for adipose tissue function per se in the observed fatty liver phenotype in Mboat7 ASO-treated mice.

Also, given the main thrust of the study is NAFLD/NASH, the authors should show collagen staining and fibrosis pathology scores.

As described above in response to reviewer # 1, we have tested a number of approaches to identify fibrosis in our mice, but under the high fat diet conditions we used we simply do not see fibrosis. In full disclosure we have made this clear to the reader. Since our original submission we have performed a number of experiments to test whether MBOAT7 knockdown indeed stimulates bridging fibrosis as is seen in advanced human liver disease, but as brought forth by this astute reviewer we failed to find pathological or biochemical support of fibrosis in this mouse model. First, we had a board-certified liver pathologist (Dr. Daniela Allende, Cleveland Clinic) blindly score H&E-, pirosirius Red, and trichrome-stained liver sections from HFD-fed control and Mboat7 ASO-treated mice, and in all cases there was no measureable fibrosis scoring above background. To indirectly quantitate fibrosis potential in a complimentary way we analyzed liver hydroxyproline content biochemically, and again found no indication that hydroxyproline levels were altered in Mboat7 ASO-treated mice. Finally, we performed some preliminary studies in primary stellate cells treating with MBOAT7 substrate lipids (LPIs) and did not see any obvious signs of pro-fibrotic responses in primary stellate cells. Given that we do not see any signs of even early stage fibrosis in Mboat7 ASO-treated mice, we have removed any discussion regarding effects on fibrosis and instead focus the discussion on the striking hepatic lipid accumulation seen with MBOAT7 loss of function in high fat fed mice. Although beyond the scope of the current study, we are now generating in parallel either hepatocyte, macrophage, or stellate cell specific Mboat7 knockout mice and will subject them to models of fibrosis induction (carbon tetrachloride, methionine/choline-deficient diet, bile duct ligation, etc.) to understand the cell autonomous effect of Mboat7 in pro-fibrotic responses.

To make this issue clear for the reader we have removed any claims that fibrosis was found in our studies, and have added the following sentences to the Discussion section:

“However, an important limitation of our studies is that high fat diet feeding in mice is not sufficient to drive bridging fibrosis. In future studies it will be important to examine how Mboat7 loss of function impacts the development of fibrosis in appropriate fibrosis-prone animal models, and whether Mboat7 expression in hepatic stellate cells plays a regulatory role the progression from NASH to cirrhosis.”

Figure 3E presents a major problem as the MBOAT7 ASO lowers mRNA for TMC4, making the rest of the experiments (most of the rest of the paper) not possible to interpret for MBOAT7 depletion. If the point of the paper is that both genes are coordinately regulated and somehow both function in these phenotypes, then more work needs to be done to understand this. However, the current paper is about MBOAT7.

We completely agree with this reviewer concern, and this was shared by several reviewers. Therefore, we felt it was imperative to address the issue with additional studies in Tmc4 deficient mice. We generated a global Tmc4 knockout mice using CRISPR-Cas9-mediated gene editing and examined hepatic lipid levels under high fat feeding conditions (Figure 4). Global Tmc4 knockout mice have marked reductions in hepatic Tmc4 mRNA (Figure 4A) and protein (Figure 4B), yet importantly Mboat7 expression is unaltered (Figure 4A). In contrast to the striking hepatic steatosis seen with Mboat7 loss of function (Figure 2E, Figure 2—figure supplement 4), Tmc4 null mice show similar levels of hepatic lipids when fed a high fat diet (Figure 4). These data strongly implicate Mboat7, but not Tmc4, as the primary mediator of hepatic steatosis seen with the rs641738 variant or in Mboat7 ASO-treated mice.

Figure 5; results for other phospholipids and lysophospholipids should be shown. Is this defect specific for PI/LPI?

To address this, we have quantified levels of most major species of phospholipids and lysophospholipids in the liver or Mboat7 ASO-treated mice, and have included these data in a new supplementary figure. Mboat7 knockdown did not significantly alter the hepatic levels of other major phospholipids including phosphatidylcholines, lysophosphatidylcholines, phosphatidylethanolamines, lysophosphatidylethanolamines, phosphatidylserines, phosphatidylglycerols, or phosphatidic acids (Figure 6—figure supplement 5). This is consistent with previously published findings from global Mboat7^-/-^ mice (Lee et al., 2012). Collectively, our data suggest that ASO-mediated knockdown of Mboat7 primarily alters LPI and PI levels in the liver and white adipose tissue.

Figure 6; it's not clear exactly how LPI is given, how much, and whether it is "physiological".

Supplementary Figure 13 shows that with our dosing regime we were able to see a rapid transient ~2-fold increase in either 18:0 LPI or 18:1 LPI. We feel that this is well within the physiological range of these lipids, given that we have found human plasma levels to be much higher than what is found in mice (Figure 7).

Reviewer #3:This is the first characterization of MBOAT7 knockdown mice. The characterization is good and complete with interesting findings. There are several important weaknesses- they don't actually establish whether MBOAT7 or TMC4 are independently or cooperatively involved in the resulting phenotype. I don't think with the experiments presented they have firmly established with the alterations in lipid species found are responsible for the inflammation and stellate cell activation. It is important they address my third major point prior to any publication as it has important implication if true. Finally, they don't address why TGs accumulate in liver. The latter seem pretty basic given this is how the region was identified in the first place. Overall, I think this is marginal for acceptance, but certainly more experiments would be required.

As described above in responses to reviewers 1 and 2, we have now added data to directly address the role of Mboat7, but not Tmc4 in hepatic steatosis. We also provide new data showing that the hepatic steatosis seen in Mboat7 ASO-treated mice is driven in part by reorganization of the proteome of cytosolic lipid droplets.

Major points:1) The effect of the ASO on FAT Mboat7 expression should be included.

Given this is a shared concern by all 3 reviewers, we have now included several additional experiments examining the effects of Mboat7 ASO treatment on adipose tissue biology. Our revised manuscript shows that despite significant knockdown of Mboat7 in adipose tissue, there are no difference in adiposity (Figure 2—figure supplement 1), basal and catecholamine-stimulated lipolysis (Figure 9c), energy expenditure (Figure 2—figure supplement 3), and only very minor differences in adipose tissue gene expression (Figure 2—figure supplement 1). Therefore, our data do not support a strong role for adipose tissue function in the observed fatty liver phenotype in Mboat7 ASO-treated mice.

2) Regarding the experimental design of Figure 6. It is not clear whether the amount of 18:0 LPI injected resulted in a 2-fold increase in the total LPI concentration or in the amount of 18:0. The later would be appropriate for these experiments. It seems to directly link the LPI and the function of MBOAT7, only the species that were altered should induce the inflammation and stellate cell activation. There is a lack of controls in this important experiment, which should include species that were not altered by MBOAT7 knockdown.

This concern was shared by reviewer # 1, so we performed additional experiments and now show data with all three potential LPI substrates (16:0, 18:0, and 18:1). Since our original submission, we have now directly injected LPI species that are altered by Mboat7 ASO treatment (18:1 LPI and 16:0) and ones that are not altered by Mboat7 ASO treatment (18:0 LPI). It is important to consider that although only 16:0 and 18:1 LPI accumulate in the liver of Mboat7 ASO-treated mice (Figure 6D), we did find reduced hepatic LPIAT activity with all three potential substrates (Figure 6A). Furthermore, cytosolic lipid droplets isolated from Mboat7 ASO-treated mice showed marked accumulation of 18:0 LPI, but not 16:0 LPI or 18:1 LPI (Figure 9D). This is in stark contrast to what is seen in the whole liver, where Mboat7 knockdown instead promotes accumulation of 16:0 LPI and 18:1 LPI (Figure 6D). Functionally, we found that injection of either 18 carbon LPI (18:0 or 18:1), but not 16:0 LPI, was able to increase pro-inflammatory and pro-fibrotic gene expression, but only in the Mboat7 ASO-treated group. These new data are shown in main Figure 8 and Figure 8—figure supplement 1.

3) What is the mechanism by which the MBOAT7 knockdown results in TG, CE and FC accumulation in liver?

As described above in response to reviewer 1, we have performed a number of additional experiments to understand the mechanism(s) by which Mboat7 knockdown promotes hepatic steatosis. First, we have repeatedly seen that Mboat7 knockdown does not significantly alter Srebp1c or lipogenic gene expression, either under fed or fasted conditions (Figure 9A). We have also consistently seen no alterations in steady state plasma triglyceride levels (Figure 2—figure supplement 5) or VLDL triglyceride secretion rates (Figure 9B). After ruling out these common causes of hepatic steatosis, we instead focused our attention on whether MBOAT7 knockdown could drive steatosis by locally altering the lipid and proteome composition of cytosolic lipid droplets (i.e. impacting lipid storage at the lipid droplet interface). The rationale for pursuing this possibility was that a recent manuscript (Mancina et al., 2016) suggested that MBOAT7 can co-purify with cytosolic lipid droplets in subcellular fractionation experiments. Using sucrose gradient lipid droplet fractionation protocols, we have now confirmed that MBOAT7 can be found on cytosolic lipid droplets in control mice but is essentially absent on lipid droplets from Mboat7 ASO-treated mice (Figure 9F). In parallel with MBOAT7 localization to lipid droplets, we found the lipid droplets isolated from Mboat7 ASO-treated mice show striking increases in MBOAT7 substrates (18:0 LPI and 20:4 LPI) and reduction in MBOAT7 products (38:4 PI). In fact, the differences seen in LPI and PI lipids are much more apparent in isolated lipid droplets (Figure 9) than what is seen in whole liver homogenates (Figure 6). Collectively, our results support a role for MBOAT7 in regulating local LPI acylation at the lipid droplet surface, which is associated with marked accumulation of lipogenic enzymes (CCTα and GPAT4) on lipid droplets, which may contribute to the large lipid droplets seen Mboat7 ASO-treated mice. These new data are discussed in the revised manuscript as follows:

“From this study it is clear that MBOAT7 can diversify the inositol-containing phospholipids and the associated proteome on cytosolic lipid droplets, and this could in part explain the large lipid droplets that accumulate in Mboat7 ASO-treated mice. It is interesting to note that the well known PNPLA3 variant associated with fatty liver disease (I148M) accumulates on lipid droplets, and similarly reorganizes the lipidome and proteome to promote liver disease progression^35-38^. Our findings here with MBOAT7-driven restructuring of the lipid droplet surface, and those recently published with PNPLA3^35-38^, suggest that alterations in lipid modifying enzyme access to the surface of cytosolic lipid droplets may be a common mechanism by which human fatty liver develops. Therefore, rigorous studies of the lipid droplet lipidome and proteome in freshly isolated human liver specimens may provide clues into new therapeutic leads for advanced liver diseases.”

[Editors' note: the author responses to the re-review follow.]

Summary:The reviewers and editors agree that the manuscript has been improved since the original submission and several issues have been clarified. However, the mechanism underlying the steatosis is still unclear. The new data regarding alterations to the liver lipid droplet proteome are interesting but do not establish whether these changes are a cause or a consequence of steatosis. Given that steatosis is the focus of the paper, the consensus is that this issue must be addressed prior to publication.

To address the mechanism underlying steatosis associated with MBOAT7 loss of function we have generated MBOAT7 deficient human hepatoma cells and performed several new metabolic tracer studies to further clarify the mechanism. MBOAT7-deficient Huh7 cells showed no significant difference in triacylglycerol hydrolysis rates, but instead showed modest increases in de novo lipogenesis (^14^C-acetate incorporation into ^14^C-triacylglycerol) and marked reductions in fatty acid oxidation (^14^C-palmitate incorporation into ^14^C-CO_2_). These new data are included in new Figure 10, and provide new information by which MBOAT7 control fatty acid metabolism and triacylglycerol storage in a cell autonomous manner.

Essential revisions:1) Direct evidence of how TG, CE and free cholesterol accumulates in Mboat7-deficient livers is not provided. A higher abundance of GPAT4 in the lipid droplet fraction was observed, but it is unclear where the acyl-CoA substrates for GPAT are coming from if there is no increase in de novo lipogenesis or uptake from circulation. The abundance of PLIN2, PLIN3, GPAT4, and CCTα on lipid droplets may merely reflect the presence of more lipid droplets. Increased abundance of GPAT4 and CCTα on lipid droplets does not explain why the livers of MBOAT7-deficient mice contain more CE and free cholesterol. mRNAs linked to lipogenesis were not measured in conditions that stimulate lipogenesis (i.e. re-feeding) but were instead measured after fasting or at the start of the light cycle.The following experiments could be performed in the given time frame to address this central issue:- Measure lipogenic mRNAs in re-fed conditions (see papers by Horton and colleagues on SREBP1-dependent lipogenesis).

When we received this original comment, we ascertained from our Institutional Animal Care and Use Committee (IACUC) that in fact this study would take us roughly 4 months to complete. For us to do the in vivo ASO studies looking at fasting/re-feeding along with radioisotope (^14^C-acetate and ^3^H-glycerol) we would need to amend our Institutional Animal Care and Use Committee (IACUC) protocol to include radiotracers that would take about 1.5 months before we could even get approval to start the experiments. In typical experiments we treat with ASO for 8 weeks to achieve good knockdown, which would put us another 2 months out. Therefore, any in vivo experiments would require about 4 months of time. To be responsive within the time frame, we instead focused our efforts on measuring rates of de novo lipogenesis rather than gene expression in hepatoma cell lines that we genetically deleted MBOAT7 (see comments below). Although the in vivo fasted and re-feeding experiments would provide another layer of information, we felt it is likely to be incremental in nature to the data we already show. It is our experience that if de novo lipogenic gene expression is different between groups of mice, this can be easily seen under the conditions we collected tissues (fasted vs. fed at the start of the light cycle).

- The rate of liver lipid synthesis can be assessed by i.p injection of radiolabeled glycerol and measurement of glycerol incorporation into lipid classes after separation by TLC as in Miyazaki et al., 2001 or Millar et al., 2006.

As discussed above, to be responsive within the time frame, we instead focused our efforts on measuring rates of de novo lipogenesis in hepatoma cell lines where we genetically deleted MBOAT7. To address this in cultured cells we generated Huh7 cells genetically lacking MBOAT7 using CRISPR-Cas9 gene editing. These cells accumulate large lipid droplets much like those seen in mouse liver in vivo, so we feel this is an appropriate model system to understand cell autonomous mechanisms by which MBOAT7 alters storage of triglyceride and cholesterol. MBOAT7-deficient Huh7 cells in fact showed modest increases in de novo lipogenesis (^14^C-acetate incorporation into ^14^Ctriacylglycerol) and marked reductions in fatty acid oxidation (^14^C-palmitate incorporation into ^14^C-CO_2_). These new data are included in new Figure 10, and provide new mechanistic information by which MBOAT7 control fatty acid metabolism and triacylglycerol storage in a cell autonomous manner.

- Alternatively, if they believe their data is consistent with an impairment in TG mobilization from LDs, then the rate of liver LD lipolysis can be assessed as in Schweiger et al., 2014. These measurements could be done in primary hepatocytes/cell culture models if that is easier to achieve.

As suggested, to test whether MBOAT7 loss-of-function is associated with impairment in triacylglycerol or cholesteryl ester mobilization, we performed pulse-chase studies in wild type or MBOAT7-deficient Huh7 cells using both ^14^C-oleic acid and ^3^H-cholesterol to label triacylglycerol and cholesteryl ester pools to steady state. Thereafter, cells were chased in the presence of the acyl-CoA synthetase inhibitor triacsin C (effectively blocking re-esterifcation of hydrolysis products). Using this method, we found that the turnover/mobilization of both triacylglycerol and cholesteryl esters was not altered in MBOAT7-deficient cells. These data are included in new Figure 10.

- de novo lipogenesis as a contributor to fat accumulation could be assessed by injecting mice with 14C-acetate (after they had been fasted O/N and re-fed for 6 hours) as in Lian et al., 2018.

Although we were unable to assess the effects of Mboat7 knockdown on de novo lipogenesis in vivo in the time frame, our studies clearly show that acetate incorporation into cellular triacylglycerol is increased in MBOAT7-deficient Huh7 cells compared to wild type Huh7 cells (see new Figure 10).

2) In the experiment in which LPI species were administered to mice, only mRNAs related to inflammation were assessed. Analysis of hepatic lipid metabolism should be included in this experiment, especially since the focus of the manuscript has shifted from fibrosis/inflammation to hepatic steatosis.

This is an excellent point, and we have now included data showing hepatic triacylglycerol levels in the LPI injection experiments in revised Figure 8I.